# Optimization of Air Conditioning Performance with Al₂O₃-SiO₂/PAG Composite Nanolubricants Using the Response Surface Method

**Nurul Nadia Mohd Zawawi** [1]**, Wan Hamzah Azmi** [1,2,*]**, Abd Aziz Mohamad Redhwan** [3]**,
Anwar Ilmar Ramadhan** [4] **and Hafiz Muhammad Ali** [5,6]

[1] Centre for Research in Advanced Fluid and Processes, Lebuhraya Tun Razak, Gambang,
    Kuantan 26300, Pahang, Malaysia
[2] Faculty of Mechanical and Automotive Engineering Technology, Universiti Malaysia Pahang,
    Pekan 26600, Pahang, Malaysia
[3] Faculty of Manufacturing Engineering Technology, TATI University College (TATIUC),
    Kemaman 24000, Terengganu, Malaysia
[4] Department of Mechanical Engineering, Faculty of Engineering, Universitas Muhammadiyah Jakarta,
    Jl. Cempaka Putih Tengah No 27, Jakarta 10510, Indonesia
[5] Mechanical Engineering Department, King Fahd University of Petroleum and Minerals,
    Dhahran 31261, Saudi Arabia
[6] Interdisciplinary Research Center for Renewable Energy and Power Systems (IRC-REPS),
    King Fahd University of Petroleum and Minerals, Dhahran 31261, Saudi Arabia
**\*** Correspondence: wanazmi2010@gmail.com; Tel.: +6-09-4246338 or +6-09-4242202

**Abstract:** A variety of operational parameters can influence the operation of an automobile air-conditioning (AAC) system. This issue is solved by using optimization techniques that can recommend the ideal parameters for the best results. To improve the performance of AAC system usings Al₂O₃-SiO₂/PAG composite nanolubricants, the response surface method (RSM) was employed. RSM was used to design the experimental work, which was based on a face composite design (FCD). The RSM quadratic models were helpful in determining the links between the input parameters and the responses. The addition of composite nanolubricants improved the overall performance of AAC systems. The parameters were optimized using the RSM's desirability approach, with the goal of increasing cooling capacity and the coefficient of performance (COP), while reducing compressor work and power consumption. The ideal parameters for the AAC system were found to be 900 rpm compressor speed, 155g refrigerant charge, and 0.019% volume concentration, with a high desirability of 81.60%. Test runs based on the optimum circumstances level were used to estimate and validate cooling capacity, compressor work, COP, and power consumption. Both predicted and measured values were in good agreement with each other. A new RSM model was successfully developed to predict the optimal conditions for AAC system performance.

**Keywords:** hybrid nanolubricants; refrigeration system; response surface method

## 1. Introduction

Optimization approaches using various methods are useful in determining the optimum parameters to achieve the desired performance. The investigation of the refrigeration system is time-consuming and costly when all experiments must be conducted. Thus, an optimization approach on refrigeration system parameters to find the optimum performance should be evaluated. The most commonly used methods for optimization are the RSM [1,2], Taguchi method [3,4], artificial neutral network (ANN) [5], multi-response optimization method [6], and regression analysis. Recently, improving and optimizing refrigeration system performance using software networks or modeling has begun receiving increasing attention from researchers [7,8]. This is plausible due to improved computer technology,

as well as the accessibility of simulation software. RSM is a mathematical and statistical method for improving, enhancing, and optimizing independent parameters in a set of experiments, as well as their interactions with response variables, in a process that allows it to enhance and optimize development [9,10]. RSM is devoted to estimating interactions and quadratic effects, and it is a solution to the multi-variable statistical method problem, providing an idea of the response surface local shape [11]. Likewise, RSM can aid in the quantitative and routine modification of elements that influence the AAC system's performance. The RSM has an advantage over the complete factorial approach in that it requires fewer tests to construct the experiment and less time to answer the objective problem. Therefore, computing resources are reduced.

Many studies have used the RSM optimization approach to determine the optimum working conditions. The surface roughness of EN31 steel was analyzed by Abhang and Hameedullah [12] using RSM. The feed rate, followed by the cutting speed and depth of the cut, had the greatest impact on surface roughness. During the turning process, RSM was utilized by Makadia and Nanavati [13] to generate a mathematical model for surface roughness. The feed rate had the largest impact on surface roughness, followed by the tool nose radius. The electrical discharge machining (EDM) process was modeled and optimized using RSM [14]. They discovered that RSM could be employed in most optimization-related tasks, and the advantage of RSM-based response parameter analysis was that each working parameter's effect on the value of the resultant response parameter could be explained individually. Table 1 provides a list of previous studies using a range of optimization approaches in various applications.

**Table 1.** Previous studies on optimization method approaches in various applications.

| Author (s) | Year | Fields/Applications/Systems | Optimization Methods |
|---|---|---|---|
| Abhang and Hameedullah [12] | 2011 | EN31 steel turning process | RSM |
| Barik and Mandel [15] | 2012 | EN31 steel turning process | RSM |
| Krishankant et al. [16] | 2012 | EN34 steel turning process | Taguchi Method |
| Makadia and Nanavati [13] | 2013 | EN31 steel turning process | RSM |
| Rao and Venkatasubbaiah [17] | 2016 | Surface roughness in CNC turning | Taguchi and ANOVA |
| Li et al. [18] | 2016 | CNC machining | Taguchi, RSM, and MOPSO |
| Costa and Garcia [7] | 2016 | Refrigeration systems | RSM |
| Parpas et al. [19] | 2017 | Refrigeration systems | RSM |
| Gangil and Pradhan [14] | 2017 | Electrical discharge machining (EDM) process | RSM |
| Parpas et al. [19] | 2017 | Air distribution and refrigeration systems | CFD/EES model |
| Belman-Flores et al. [20] | 2017 | Refrigeration systems | ANN |
| Nataraj et al. [21] | 2018 | CNC turning | RSM |
| Ocholi et al. [22] | 2018 | Sesame biolubricant pilot plant | RSM |
| Mao et al. [23] | 2018 | Resident air-conditioning (TAC) systems | RSM |
| Redhwan et al. [24] | 2018 | AAC systems | RSM |
| Qader et al. [25] | 2018 | Solar air heaters | RSM |
| Zendehboudi et al. [26] | 2019 | VCRS | RSM |
| Canbolat et al. [27] | 2019 | Absorption refrigeration systems | Taguchi and ANOVA |
| Zaman [28] | 2019 | Photonic radiative coolers | Taguchi |
| Vyas et al. [29] | 2019 | Capacity of lead acid battery | Taguchi |
| Huirem and Sahoo [30] | 2020 | Solar-Assisted Vapor Absorption Refrigeration Systems (SAVARS) | RSM |
| Ahmed et al. [8] | 2021 | Refrigeration systems | Multiple Methods |
| Zawawi et al. [31] | 2022 | Automotive air-conditioning Systems | Taguchi |

Software solutions for optimization techniques have been used to optimize the properties of nanolubricants [32–34], vapor compression refrigeration systems (VCRS) [26,35–39], and AAC systems [40]. Artificial intelligence approaches for modeling and optimizing refrigeration systems were evaluated by Ahmed et al. [8]. They discovered that the COP is the most important cost function to optimize, followed by overall cost, energy consumption,

and cooling capacity, according to trend analysis. To date, there are previous studies available which employ RSM approaches in order to optimize refrigeration [7,26,30] and AAC system performance [41]. Costa and Garcia [7] optimized the parameters of the refrigeration system using RSM. Parameters such as the temperature and flow rate of evaporators and condensers were considered. The behavior of R450A in VCRS was investigated by Zendehboudi et al. [26] using modeling and multi-objective optimization. They used RSM's central composite design (CCD) to calculate the impact of each variable, model the system, and develop cost functions. The compressor's power consumption was lowered by 18.39%, the discharge temperature was increased by 53.31%, and the refrigerant mass flow rate was increased by 215.57%. Huirem and Sahoo [30] used a combined Box–Behnken statistical design (BBD) and RSM technique to maximize the COP, exergetic COP (ECOP), and total exergy destruction (TED) of a LiBr-$H_2O$ vapor absorption refrigeration system.

A concept of using two or more metal oxide nanoparticles in existing lubricants—known as composite nanolubricants—is adapted due to the limited contribution of single nanolubricants in terms of the stability, compressor operations, wear rates, and performance of AAC system. Nanofluids/nanolubricants have distinct thermal physical and tribological properties, as well as performance, compared to base fluids, according to several investigations [42–45]. Previous studies on the thermal physical and tribological properties, along with the performance and optimization of AAC using PAG based single-component nanolubricants with $SiO_2$, $Al_2O_3$, and $TiO_2$ metal oxides, are available in the literature [46–50]. Zawawi et al. [51] examined the thermal conductivity of single $Al_2O_3$, $SiO_2$, and metal oxide composite nanolubricants. Based on the comparison, metal oxide composite nanolubricants have a substantially higher thermal conductivity than single nanolubricants. Additionally, few studies investigated the performance of single nanolubricants and composite nanolubricants in refrigeration and AAC systems [41,52,53]. Sharif et al. [53] examined the performance of the AAC system employing $SiO_2$/PAG nanolubricants. They discovered a maximal COP enhancement of up to 24%. In another study, Redhwan et al. [41] claimed that COP and cooling capacity were improved by up to 31% and 32%, respectively, in another experiment. Meanwhile, Zawawi et al. [54] found that $Al_2O_3$-$SiO_2$/PAG composite nanolubricants showed greater COP and cooling capacity increases than single nanolubricants, with values of 28.10% and 65.21%, respectively, at 0.015% volume concentration. For the optimization of nanolubricants, Redhwan et al. [24] used the RSM approach to study the AAC system performance using single-component $Al_2O_3$ nanolubricants in a PAG based nanolubricant in AAC systems. They found that the compressor speed, initial refrigerant charge, and nanolubricant volume concentration all have a significant impact on the AAC system's efficiency. The literature on the use of composite nanolubricants to improve the performance of AAC systems is scarce [55]. Despite this, no further research into the performance improvement of AAC systems using composite nanolubricants by employing RSM has been done in recent years.

Previous studies have reported on the impact of single-component nanolubricants on refrigeration and AAC system performance; however, more research into the effects of AAC system parameters operating with $Al_2O_3$-$SiO_2$/PAG using RSM is still essential. Therefore, in this study, the effects of operational parameters on COP, cooling capacity, compressor work ($W_{in}$), and power consumption for $Al_2O_3$-$SiO_2$/PAG nanolubricants in AAC systems were explored using RSM. The current study makes use of Design–Expert software, and the experiments are designed employing the FCD procedure. For maximum augmentation in COP and cooling capacity, as well as maximum decrease in $W_{in}$ and power consumption, optimal operating settings, such as speed, initial refrigerant charge, and composite nanolubricants volume concentration, were determined.

## 2. Materials and Methods

### 2.1. Preparation of $Al_2O_3$-$SiO_2$ Composite Nanolubricants

In this investigation, $Al_2O_3$ and $SiO_2$ nanoparticles in dry powder form, as well as polyalkylene glycol (PAG) 46, were employed. Table 2 lists the features of these

nanoparticles [46,56], and Table 3 illustrates the characteristics of PAG 46 lubricant at atmospheric pressure [57]. To confirm the existence of the nanoparticles, a chemical composition test was performed. The chemical composition of both nanoparticles was assessed by EDX analysis, as shown in Figure 1. In Figure 1a,b, the elemental composition of the materials for $Al_2O_3$ and $SiO_2$ nanoparticles, respectively, was validated. TEM evaluation was carried out for the composite nanolubricant to observe the colloidal nanoparticle dispersion in nanolubricants. Figure 2 shows TEM imaging of the $Al_2O_3$-$SiO_2$/PAG composite nanolubricants. Both nanoparticles were discovered to be spherical. In addition, the graph demonstrates the presence of two groups of nanoparticles with various diameters. $Al_2O_3$ nanoparticles are represented by the smaller diameter particles, while $SiO_2$ nanoparticles are represented by the larger diameter particles. The appearance of nanoparticles in grayscale shades may be caused by overlap particles and small aggregation. The formulation and characterization of composite nanolubricants was previously addressed in the literature. Therefore, this study focused on the preparation and formulation procedures for composite nanolubricants.

**Table 2.** Properties of nanoparticles [46,56].

| Properties | $Al_2O_3$ | $SiO_2$ |
|---|---|---|
| Molecular mass (g/mol) | 101.96 | 60.08 |
| Average particle diameter (nm) | 13 | 30 |
| Density (kg/m$^3$) | 4000 | 2220 |
| Thermal Conductivity (W/m.k) | 36 | 1.4 |
| Specific heat (J/kg·K) | 773 | 745 |

**Table 3.** Properties of PAG 46 lubricant [57].

| Properties | PAG 46 |
|---|---|
| Density, g/cm$^3$ @ 20 °C | 0.9954 |
| Flash Point, °C | 174 |
| Kinematic viscosity, cSt @ 40 °C | 41.4–50.6 |
| Pour point, °C | −51 |

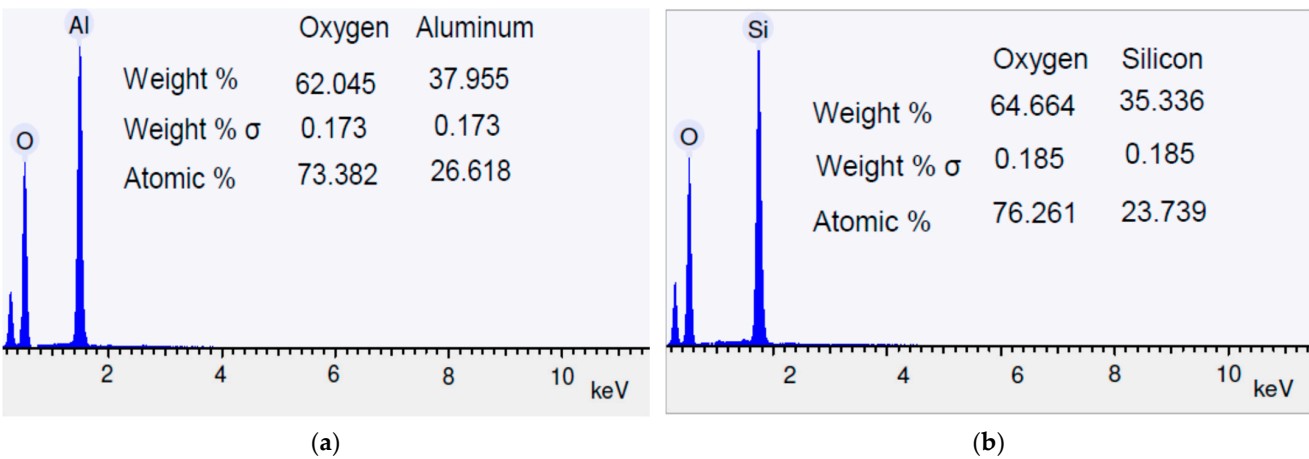

(**a**)                   (**b**)

**Figure 1.** The elemental composition of the nanoparticles (**a**) $Al_2O_3$; (**b**) $SiO_2$.

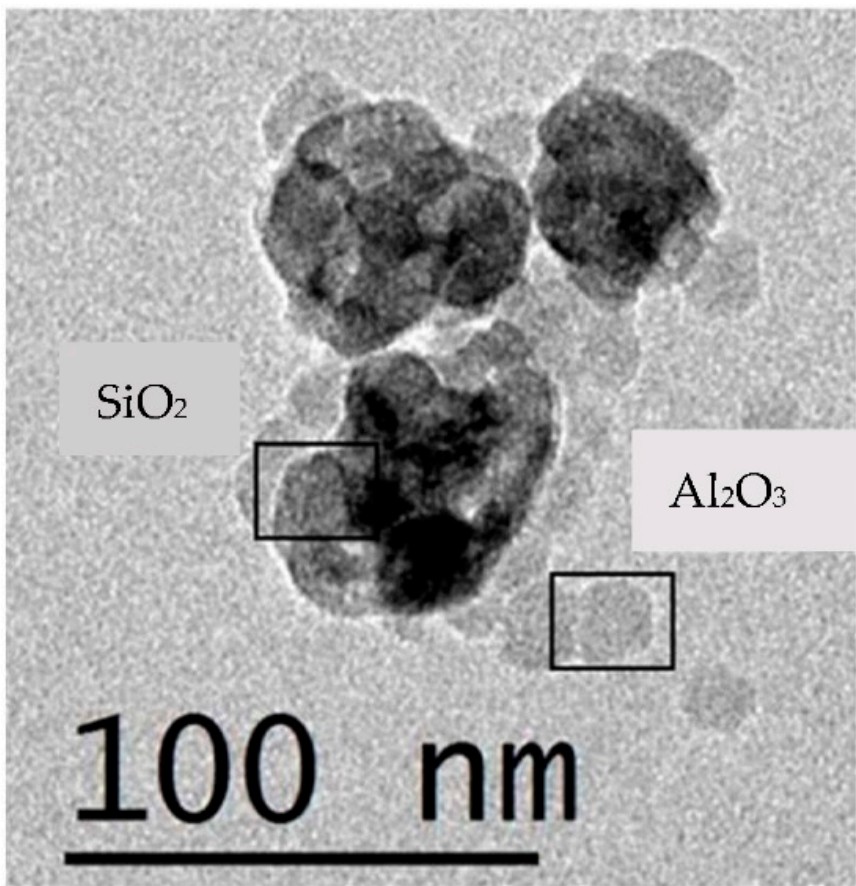

**Figure 2.** TEM image of composite nanolubricants.

In this study, the $Al_2O_3$-$SiO_2$/PAG composite nanolubricants were made utilizing a two-step procedure, and their stability was then investigated using UV-Vis and zeta potential. Zawawi et al. [51] found that the best combination for both nanolubricants used is a composition ratio of 60:40. The $Al_2O_3$-$SiO_2$/PAG composite nanolubricants in a 60:40 ratio, according to the authors, produces better thermal characteristics [58], tribological behavior [59], and AAC system performance [54] compared to other combination ratios. Therefore, the optimum ratio for $Al_2O_3$-$SiO_2$/PAG composite nanolubricants is chosen for the current work as a continuation of the prior work. The nanolubricants of $Al_2O_3$/PAG and $SiO_2$/PAG were first prepared separately. A total volume of 63 mL of $Al_2O_3$/PAG nanolubricants was then mixed with 42 mL $SiO_2$/PAG using a magnetic stirrer. The desired volume concentrations used in this study are 0.005% up to 0.045%. Equation (1) is used to compute the volume concentration of the composite nanolubricants.

$$\phi = \frac{m_p/\rho_p}{m_p/\rho_p + m_L/\rho_L} \times 100 \qquad (1)$$

where $\phi$ is the volume concentration of nanolubricants (%), $m_p$ is the nanoparticle mass (g), $\rho_p$ is the nanoparticle density ($kg/m^3$), $m_L$ is the lubricant mass (g), and $\rho_p$ is the lubricant density ($kg/m^3$). The prepared composite nanolubricants were then sonicated in an ultrasonic bath for 2 h for a uniform dispersion and stable suspension, based on previous works [51,55,58,60–62], and shown in the Figure 3. The absorbance ratio of the mixed nanolubricant dispersions, measured at various sonication durations (0 to 2.0 h) up to 700 h, is shown in Figure 3. The graph is used to determine the ideal sonication duration required to preserve the stability of $Al_2O_3$-$SiO_2$/PAG composite nanolubricants. With the most stable composite nanolubricants, the absorbance ratio with the highest value indicates

the ideal sonication time. According to the graph, two hours of sonication sustained the concentration ratio beyond 90%, even after up to 700 h of sedimentation.

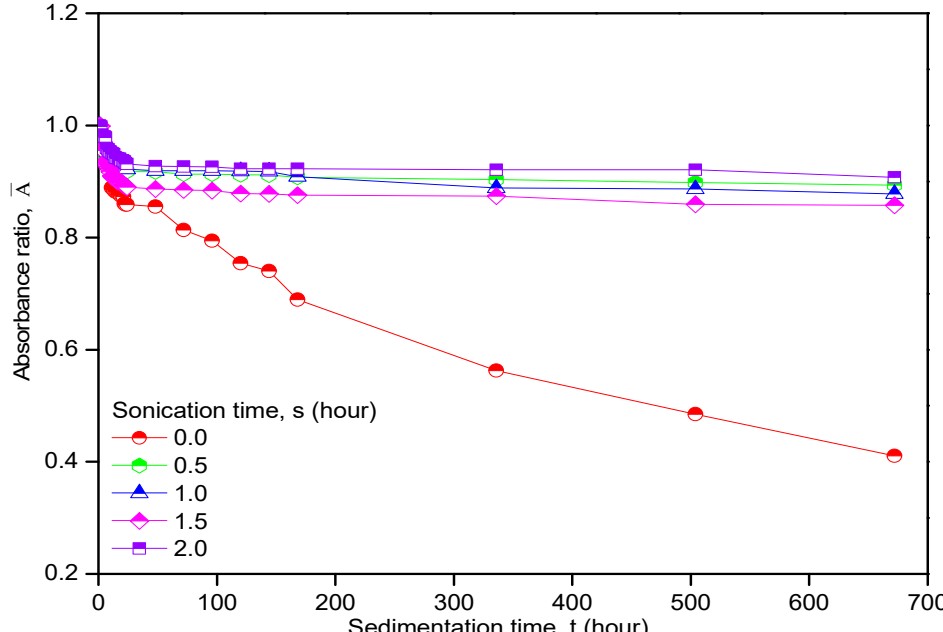

**Figure 3.** Composite nanolubricant with various sonication times.

The zeta potential and zeta sizer were used in the study to analyze the zeta potential reading and polydispersity index (PDI) of the composite nanolubricants. The current absolute zeta potential reading for the $Al_2O_3$-$SiO_2$/PAG is up to 61.1 mV. The zeta potential for $Al_2O_3$-$SiO_2$/PAG was found to be beyond the stable limit, thus proving an excellent stability. The current absolute zeta potential reading for the $Al_2O_3$-$SiO_2$/PAG is up to 61.1 mV, whereas other combination of metal oxides, i.e., $Al_2O_3$-$TiO_2$/PAG and $TiO_2$-$SiO_2$/PAG composite nanolubricants, which were studied prior to this work [51], recorded up to 31.7 mV and 22.7 mV, respectively. Previously, Redhwan et al. [41] reported that the zeta potential for $Al_2O_3$/PAG single nanolubricants was 37.8 mV. When compared to single-component nanolubricants, the $Al_2O_3$-$SiO_2$/PAG composite nanolubricants employed in this study showed improved stability. The present results were compared to the stability classification suggested by Lee et al. [63], as shown in Figure 4. The zeta potential for $Al_2O_3$-$SiO_2$/PAG was found to be beyond the stable limit, thus proving an excellent stability. The breadth or spread of the particle size distribution is described by the PDI, which is another crucial metric [64]. The maximum PDI value was found to be 0.86 for the $Al_2O_3$-$TiO_2$/PAG composite nanolubricants, while the lowest PDI value was found to be 0.22 for the $Al_2O_3$-$SiO_2$/PAG, as can be seen in Figure 4. In light of this, it should be observed that the lowest PDI value is quite comparable to that of the monodisperse state. A suspension will be monodisperse, according to Sadeghi et al. [65], if the PDI value is less than 0.3, and the size distribution curve has a single peak.

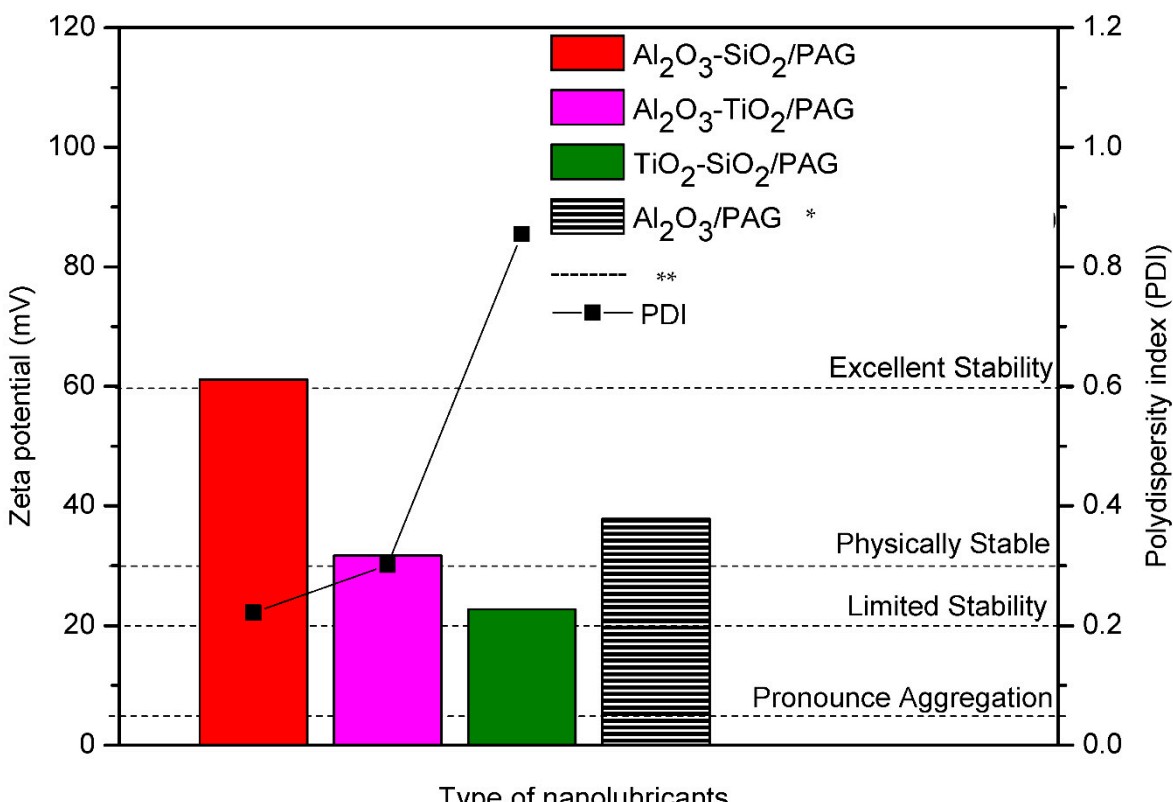

**Figure 4.** Zeta potential measurement and polydispersity index (PDI). * Redhwan et al. [41].
** Lee et al. [63].

### 2.2. Design of Experiment with RSM

The initial step in RSM is to confirm a range with the optimal condition. Secondly, the relationship model between response and the group of independent factors must be established. The last stage is to optimize the process with the model. A selection of elements in RSM study included batch tests on AAC performance with parameters of composite nanolubricants including volume concentrations, compressor speeds, and refrigerant charges. Meanwhile, cooling capacity, compressor work, COP, and power consumption were selected for the output responses of the experiment. The RSM is used to optimize all AAC system performance responses simultaneously by incorporating them into a single objective function. The objective of RSM in the current study is to examine the effect of the compressor speed, initial refrigerant charge, and volume concentration of composite nanolubricants on the AAC system performance.

The CCD was used to optimize the model, and it worked well for fitting a quadratic surface and for process optimization in general. In this study, FCD is used because there is a common area of interest and operability, and the trials are based on the design matrix. Each parameter includes three levels of variation: (i) high (+1), (ii) low (−1), and (iii) center points (coded as level 0). Six central points, six axial points, and eight factorial points were used in this study, with alpha $\alpha = 1$. The $\alpha$ value is denoted as the distance between each axial point and the CCD's center [24]. Multi-objective responses of AAC performance optimization of optimum design, with the highest desirability, are sought. Three AAC system parameters, with their levels according to RSM analysis, were investigated. Twenty experimental runs, including six replicates at the center point, were used in an FCD with three factors and three levels. The factor levels of the independent variables for AAC system performance were previously shown in Table 4. Table 5 illustrates the complete design matrix for the experiments to be conducted, as well as the collected findings, which were

analysed using analysis of variance (ANOVA) by Design–Expert Software (V13, Stat-Ease Inc., Minneapolis, MN, USA).

**Table 4.** AAC system design parameter.

| Level | A-Volume Concentration, $\varphi$ (%) | B-Compressor Speed (rpm) | C-Refrigerant Charge (g) |
|---|---|---|---|
| −1 | 0.005 | 900 | 95 |
| 0 | 0.025 | 1500 | 125 |
| 1 | 0.045 | 2100 | 155 |

**Table 5.** The design of the experiment (DOE) and the results from the experiments.

| $\varphi$ (%) | Speed (rpm) | Refrigerant Charge (g) | Cooling Capacity (kW) | Compressor Work (kJ/kg) | COP | Power Consumption (kW) |
|---|---|---|---|---|---|---|
| 0.005 | 900 | 95 | 0.665 | 23.10 | 8.13 | 0.61 |
| 0.045 | 900 | 95 | 0.477 | 24.80 | 7.65 | 0.59 |
| 0.005 | 2100 | 95 | 0.860 | 39.20 | 4.72 | 1.07 |
| 0.045 | 2100 | 95 | 0.568 | 43.10 | 4.31 | 1.06 |
| 0.005 | 900 | 155 | 0.777 | 19.70 | 9.16 | 0.68 |
| 0.045 | 900 | 155 | 0.873 | 20.20 | 8.66 | 0.73 |
| 0.005 | 2100 | 155 | 1.452 | 32.20 | 5.15 | 1.42 |
| 0.045 | 2100 | 155 | 0.954 | 34.50 | 4.87 | 1.34 |
| 0.005 | 1500 | 125 | 0.956 | 32.80 | 6.06 | 0.94 |
| 0.045 | 1500 | 125 | 0.770 | 33.30 | 5.62 | 0.89 |
| 0.025 | 900 | 125 | 0.797 | 21.90 | 8.52 | 0.60 |
| 0.025 | 2100 | 125 | 0.891 | 37.35 | 4.81 | 1.08 |
| 0.025 | 1500 | 95 | 0.667 | 33.00 | 5.49 | 0.71 |
| 0.025 | 1500 | 155 | 1.168 | 26.60 | 6.27 | 0.89 |
| 0.025 | 1500 | 125 | 0.832 | 31.00 | 5.85 | 0.85 |
| 0.025 | 1500 | 125 | 0.832 | 31.00 | 5.85 | 0.85 |
| 0.025 | 1500 | 125 | 0.832 | 31.00 | 5.85 | 0.85 |
| 0.025 | 1500 | 125 | 0.832 | 31.00 | 5.85 | 0.85 |
| 0.025 | 1500 | 125 | 0.832 | 31.00 | 5.85 | 0.85 |
| 0.025 | 1500 | 125 | 0.832 | 31.00 | 5.85 | 0.85 |

*2.3. Data Analysis Using RSM*

The model's adequacy is further determined using ANOVA. The significance of each term in the model equation is used to calculate the goodness of fit in each case. The data is subjected to regression analysis to obtain the coefficient of the regression equation. Three-dimensional surface plots are then generated from the validated models. The normal plot of the residuals, predicted against the actual plots for all responses, were presented to ensure that the chosen model was suitable for predicting the response variables in the experimental values. Good agreement of both values is important for verification of the model [25]. The distribution of the close points along the straight lines indicates a good agreement between the test values and the calculated response values [22]. The normal probability is plotted to check for the residual range. Response surface plots as a function of two independent variables or factors, with the other parameters held constant, are useful tools for evaluating the interaction and correlation of the variables, as well as comprehending the main and interactive effects [66,67]. These surface plots are used to locate the optimum points of operating parameters to attain maximum performance of the AAC system. The desirability technique of RSM can ultimately be used to find the best combination of speed, refrigerant charge, and volume concentration of composite nanolubricants.

### 3. Results and Discussion

#### 3.1. ANOVA Analysis

A summary of $p$-value and model statistics for cooling capacity are shown in Table 6. The CCD module suggested that a linear and 2FI model to be use for analysis. In order to analyse the cooling capability, linear and two-way interaction (2FI) models were both employed. The model has been improved by the addition of linear and interaction components, as shown by the low $p$-value (Prob > F). The quadratic model is not suggested for this case. The Qubic model was noted as aliased because of the existence of aliased terms. The Qubic model was not suggested, due to the insufficient running of experiments to independently estimate all the terms. Table 7 shows ANOVA analysis for cooling capacity. The model F value is noted at 25.16. This indicates that the proposed model is significant. A 95% significant level was used throughout all response analyses. Model terms with $p$-values less than 0.05 are considered as significant. The model terms are not significant if the value is larger than 0.10. All terms except AC, which is the combination of volume concentration and refrigerant charge, were significant. The fitness of model equation is validated by referring to the coefficient of regression $R^2$. $R^2$ = 92.07% for cooling capacity, demonstrating that the model could accurately predict the response. The closer the $R^2$ value to 1, the better the models fits the experimental data [68]. Pred $R^2$ of 0.3576 showed a great difference to the adj $R^2$ of 0.8841. Thus, model reduction was suggested. The signal-to-noise ratio is measured by Adeq precision, and a ratio greater than 4 is desired [22]. The signal was adequate in this case, with a ratio of 21.986.

**Table 6.** $P$-value and model summary statistics for cooling capacity.

| Source | $p$-Value | Std. Dev | $R^2$ | Adj $R^2$ (%) | Pred $R^2$ (%) | Remark |
|---|---|---|---|---|---|---|
| Linear | <0.0001 | 0.097 | 0.8075 | 0.7714 | 0.5784 | suggested |
| 2FI | 0.0076 | 0.069 | 0.9207 | 0.8841 | 0.3576 | suggested |
| Quadratic | 0.7544 | 0.075 | 0.9293 | 0.8656 | −0.1099 | not suggested |
| Qubic | 0.0088 | 0.035 | 0.9905 | 0.9698 | −10.7188 | aliased |

**Table 7.** ANOVA response for cooling capacity.

| Source | Sum of Squares | df | Mean Square | F Value | $p$-Value | |
|---|---|---|---|---|---|---|
| Model | 0.70 | 6 | 0.17 | 25.16 | <0.0001 | significant |
| A | 0.11 | 1 | 0.11 | 23.62 | 0.0003 | |
| B | 0.13 | 1 | 0.13 | 26.84 | 0.0002 | |
| C | 0.39 | 1 | 0.39 | 81.94 | <0.0001 | |
| AB | 0.061 | 1 | 0.061 | 12.66 | 0.0035 | |
| AC | $7.849 \times 10^{-4}$ | 1 | $7.849 \times 10^{-4}$ | 0.16 | 0.6931 | |
| BC | 0.028 | 1 | 0.028 | 5.7 | 0.0325 | |
| Residual | 0.063 | 13 | $4.819 \times 10^{-3}$ | | | |
| Lack of fit | 0.063 | 8 | $7.830 \times 10^{-3}$ | | | |
| Pure error | 0.000 | 5 | 0.000 | | | |
| $R^2$ | | | | | | 0.9207 |
| Adj $R^2$ | | | | | | 0.8841 |
| Pred $R^2$ | | | | | | 0.3576 |
| Adeq Precision | | | | | | 21.986 |

Table 8 represents $p$-value and model summary statistics for compressor work. The CCD module suggested that a quadratic model be use for analysis. The Qubic model was not suggested for this case. The ANOVA analysis for compressor work was recorded in Table 9. The model F value = 536.88 implicated that the model was significant. All terms were significant. The fitness of the model equation is validated by referring to the coefficient of regression $R^2$. For compressor work, with an $R^2$ of 99.79%, the model was able

to accurately predict the response. The Pred $R^2$ of 0.9852 was in reasonable agreement with the adj $R^2$ of 0.9961. An adequate signal was confirmed by the Adeq precision of 84.751.

**Table 8.** *P*-value and model summary statistics for compressor work.

| Source | *p*-Value | Std. Dev | $R^2$ | Adj $R^2$ (%) | Pred $R^2$ (%) | Remark |
|---|---|---|---|---|---|---|
| Linear | <0.0001 | 1.47 | 0.9520 | 0.9429 | 0.9133 | not suggested |
| 2FI | 0.1947 | 1.37 | 0.9661 | 0.9505 | 0.8683 | not suggested |
| Quadratic | <0.0001 | 0.39 | 0.9979 | 0.9961 | 0.9852 | suggested |
| Qubic | 0.0377 | 0.24 | 0.9995 | 0.9985 | 0.4299 | aliased |

**Table 9.** ANOVA response for compressor work.

| Source | Sum of Squares | df | Mean Square | F Value | *p*-Value | |
|---|---|---|---|---|---|---|
| Model | 718.55 | 9 | 79.84 | 536.88 | <0.0001 | significant |
| A | 7.92 | 1 | 7.92 | 53.27 | <0.0001 | |
| B | 587.52 | 1 | 587.52 | 3950.82 | <0.0001 | |
| C | 90.00 | 1 | 90.00 | 605.21 | <0.0001 | |
| AB | 2.00 | 1 | 2.00 | 13.45 | 0.0043 | |
| AC | 0.98 | 1 | 0.98 | 6.59 | 0.0280 | |
| BC | 7.22 | 1 | 7.22 | 48.55 | <0.0001 | |
| $A^2$ | 8.25 | 1 | 8.25 | 55.46 | <0.0001 | |
| $B^2$ | 7.88 | 1 | 7.88 | 53.02 | <0.0001 | |
| $C^2$ | 6.34 | 1 | 6.34 | 42.62 | <0.0001 | |
| Residual | 1.49 | 10 | 0.15 | | | |
| Lack of fit | 1.49 | 5 | 0.30 | | | |
| Pure error | 0.000 | 5 | 0.000 | | | |
| $R^2$ | | | | | | 0.9979 |
| Adj $R^2$ | | | | | | 0.9961 |
| Pred $R^2$ | | | | | | 0.9822 |
| Adeq Precision | | | | | | 84.751 |

Table 10 represents *p*-value and model summary statistics for COP. The CCD module suggested that a quadratic model be use for analysis. The Qubic model was not suggested for this case. The ANOVA analysis for COP was recorded in Table 11. The model F value = 4604.92 implicated that the model was significant. Only the combination of A (volume concentration), B (compressor speed), C (refrigerant charge), and between the AB, BC, $A^2$ and $B^2$ terms, were considered significant. Thus, all insignificant terms were eliminated. $R^2$ = 99.98% for COP, indicating that the model was capable of accurately predicting the response. The adj $R^2$ of 0.9974 and the Pred $R^2$ of 0.9995 were in reasonable agreement. A signal with a precision of 225.476 was considered adequate.

**Table 10.** *P*-value and model summary statistics for COP.

| Source | *p*-Value | Std. Dev | $R^2$ | Adj $R^2$ (%) | Pred $R^2$ (%) | Remark |
|---|---|---|---|---|---|---|
| Linear | <0.0001 | 0.43 | 0.9236 | 0.9093 | 0.8706 | not suggested |
| 2FI | 0.8700 | 0.46 | 0.9276 | 0.8941 | 0.5939 | not suggested |
| Quadratic | <0.0001 | 0.030 | 0.9998 | 0.9995 | 0.9974 | suggested |
| Qubic | 0.1951 | 0.025 | 0.9999 | 0.9997 | 0.8774 | aliased |

The summary of *p*-value and model statistics for power consumption are shown in Table 12. The CCD module suggested that a linear and quadratic model be used for analysis. Due to their superior accuracy over linear models, quadratic models were chosen. The Qubic model was noted as aliased and was not suggested due to insufficient running experiments to independently estimate all the terms. Table 13 shows the ANOVA analysis for power consumption. The F value for the model is 151.49, implying that the model is

adequate. A 95% significant level was used throughout all response analyses. Model terms with $p$-values less than 0.05 are considered significant. All values greater than 0.10, on the other hand, imply that the model terms are not significant. In this case A, B, C, BC, and $A^2$ are significant model terms. The fitness of the model equation is validated by referring to the coefficient of regression $R^2$. The model was able to predict the reaction with a high accuracy for power consumption, with $R^2 = 99.27\%$. Pred $R^2$ of 0.9862 showed a great difference compared to the Adj $R^2$ of 0.9181. Thus, model reduction was suggested. Adeq presicion was noted at 41.908, which indicated an adequate signal model.

**Table 11.** ANOVA response for COP.

| Source | Sum of Squares | df | Mean Square | F Value | $p$-Value | |
|---|---|---|---|---|---|---|
| Model | 38.21 | 9 | 4.25 | 4604.92 | <0.0001 | significant |
| A | 0.45 | 1 | 0.45 | 489.34 | <0.0001 | |
| B | 33.40 | 1 | 33.40 | 36228.04 | <0.0001 | |
| C | 1.45 | 1 | 1.45 | 1570.68 | <0.0001 | |
| AB | 0.010 | 1 | 0.010 | 11.22 | 0.0074 | |
| AC | $1.886 \times 10^{-3}$ | 1 | $1.886 \times 10^{-3}$ | 2.05 | 0.1831 | |
| BC | 0.14 | 1 | 0.14 | 149.91 | <0.0001 | |
| $A^2$ | $6.006 \times 10^{-3}$ | 1 | $6.006 \times 10^{-3}$ | 6.51 | 0.0288 | |
| $B^2$ | 1.66 | 1 | 1.66 | 1798.26 | <0.0001 | |
| $C^2$ | $3.415 \times 10^{-4}$ | 1 | $3.415 \times 10^{-4}$ | 0.37 | 0.5564 | |
| Residual | $9.220 \times 10^{-3}$ | 10 | $9.220 \times 10^{-4}$ | | | |
| Lack of fit | $9.220 \times 10^{-3}$ | 5 | $1.844 \times 10^{-3}$ | | | |
| Pure error | 0.000 | 5 | 0.000 | | | |
| $R^2$ | | | | | 0.9998 | |
| Adj $R^2$ | | | | | 0.9995 | |
| Pred $R^2$ | | | | | 0.9974 | |
| Adeq Precision | | | | | 225.476 | |

**Table 12.** *P*-value and model summary statistics for power consumption.

| Source | $p$-Value | Std. Dev | $R^2$ | Adj $R^2$ (%) | Pred $R^2$ (%) | Remark |
|---|---|---|---|---|---|---|
| Linear | <0.0001 | 0.024 | 0.9556 | 0.9473 | 0.9230 | suggested |
| 2FI | 0.5746 | 0.025 | 0.9617 | 0.9440 | 0.8285 | not suggested |
| Quadratic | 0.0006 | 0.012 | 0.9927 | 0.9862 | 0.9181 | suggested |
| Qubic | 0.3124 | 0.011 | 0.9964 | 0.9885 | −3.4727 | aliased |

### 3.2. Regression Analysis

Regression analysis was used to fit the supplied RSM response to a quadratic equation, to analyse the link between the inputs and outputs of the models, and to determine the ideal input parameters [25]. All insignificant terms are deleted to reduce the regression model. For cooling capacity, only A, B, C, and AB are chosen as significant model terms. Meanwhile, for compressor work, all terms are significant model terms. Significant model terms A, B, C, AB, BC, $A^2$, and $B^2$ were selected for COP, and A, B, C, BC, and $A^2$ were chosen for power consumption. The difference between Pred $R^2$ and the Adj $R^2$ of less than 0.2 was desired [24]. The final equation in terms of coded factors can be completed after removing insignificant terms, as shown in Equations (2)–(5) as follows:

$$\text{Cooling capacity} = 0.84 - 0.11\,A + 0.11\,B + 0.20\,C - 0.087\,AB \tag{2}$$

$$\text{Compressor work} = 31.13 + 0.89A + 7.66B - 3.00C + 0.50AB - 0.35AC - 0.95BC + 1.73A2 + 1.73A^2 - 1.69B^2 - 1.52C^2 \tag{3}$$

$$\text{COP} = 5.87 - 0.21A - 1.83B + 0.38C + 0.036AB - 0.13BC - 0.051A^2 + 0.77B^2 \tag{4}$$

$$\text{Power consumption} = 0.84 - (9.041 \times 10^{-0.03})A + 0.28B + 0.10C + 0.052BC + 0.096A^2 \tag{5}$$

where A is the volume concentration of the composite nanolubricants (%), B is the speed (rpm), and C is the refrigerant charge (g).

**Table 13.** ANOVA response for power consumption.

| Source | Sum of Squares | df | Mean Square | F Value | *p*-Value | |
|---|---|---|---|---|---|---|
| Model | 0.21 | 9 | 0.023 | 151.49 | <0.0001 | significant |
| A | $6.882 \times 10^{-5}$ | 1 | $6.882 \times 10^{-5}$ | 0.45 | 0.005179 | |
| B | 0.18 | 1 | 0.18 | 1176.19 | <0.0001 | |
| C | 0.021 | 1 | 0.021 | 135.83 | <0.0001 | |
| AB | $2.977 \times 10^{-4}$ | 1 | $2.977 \times 10^{-4}$ | 1.94 | 0.1935 | |
| AC | $5.249 \times 10^{-5}$ | 1 | $5.249 \times 10^{-5}$ | 0.34 | 0.5713 | |
| BC | $9.312 \times 10^{-4}$ | 1 | $9.312 \times 10^{-4}$ | 6.08 | 0.0334 | |
| $A^2$ | $5.971 \times 10^{-3}$ | 1 | $5.971 \times 10^{-3}$ | 38.97 | <0.0001 | |
| $B^2$ | $2.481 \times 10^{-4}$ | 1 | $2.481 \times 10^{-4}$ | 1.62 | 0.2320 | |
| $C^2$ | $5.313 \times 10^{-4}$ | 1 | $5.313 \times 10^{-4}$ | 3.47 | 0.0922 | |
| Residual | $1.532 \times 10^{-3}$ | 10 | $1.532 \times 10^{-4}$ | $1.532 \times 10^{-3}$ | | |
| Lack of fit | $1.532 \times 10^{-3}$ | 5 | $3.064 \times 10^{-4}$ | $1.532 \times 10^{-3}$ | | |
| Pure error | 0.000 | 5 | 0.000 | 0.000 | | |
| $R^2$ | | | | | | 0.9927 |
| Adj $R^2$ | | | | | | 0.9862 |
| Pred $R^2$ | | | | | | 0.9181 |
| Adeq precision | | | | | | 41.908 |

### 3.3. Residual and Response Surface Plots

Figures 5 and 6 depict a normal plot of residuals, as well as the normal plot projected against the actual plots for all responses. To compare the two values and examine the distribution of the residuals, the predicted and actual values of cooling capacity, compressor work, COP, and power consumption were plotted. All residuals in the graphs are located on a straight line, indicating that the errors have a normal distribution. The normal probability plot for any ANOVA should be evaluated for the range of residuals near the mean line, showing that residuals are generally fitted for all responses. Therefore, it can be concluded that the model for predicting AAC performance using RSM's design factors, when applied to a specific set of parameters, has a high level of accuracy.

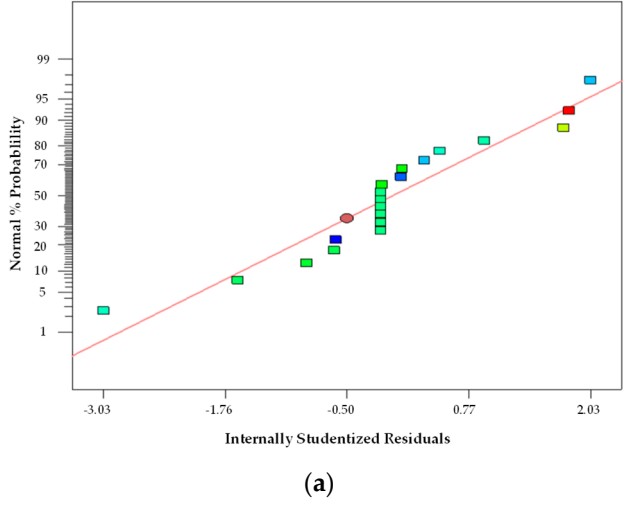

(a)

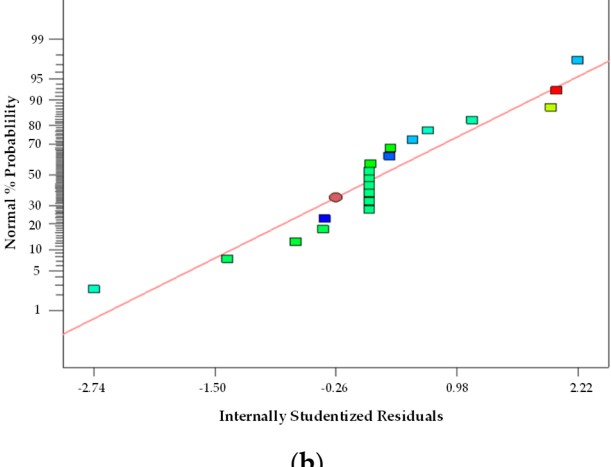

(b)

**Figure 5.** *Cont.*

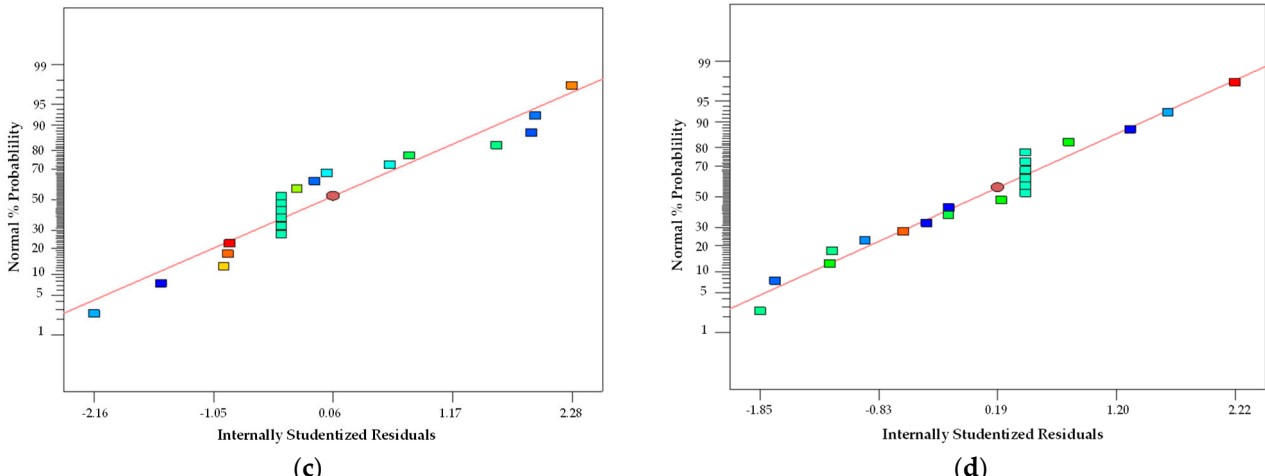

**Figure 5.** Normal plot of residuals: (**a**) cooling capacity; (**b**) compressor work; (**c**) COP; (**d**) power consumption.

**(a)**

**(b)**

**(c)**

**(d)**

**Figure 6.** Comparison of numerical and predicted values of the RSM model: (**a**) cooling capacity; (**b**) compressor work; (**c**) COP; (**d**) power consumption.

Response surface plots as a function of two independent variables or factors, with the other parameters held constant, are useful tools for evaluating the interaction and correlation of the variables, as well as for comprehending their main and interactive

effects [66,67]. Figure 7 represents the interaction of volume concentration (0.005 to 0.045%) and speed (900 to 2100 rpm) and its effect on cooling capacity when the refrigerant charge is kept constant at 125 g. Increasing volume concentrations with increment of speed reduced the cooling capacity rate. Compressor speed has a greater effect on cooling capacity, as shown through the comparison of the slope between the volume concentration of composite nanolubricants and speed. Figure 8 shows the variation of the compressor work with speed for different refrigerant charges, while volume concentration is fixed at 0.025%. From the figure, compressor work increases against the increasing speed, but decreases with refrigerant charge. The interaction of volume concentration and speed on COP is shown in Figure 9. Refrigerant charge was fixed at 125 g. It was observed that volume concentration had a greater influence on COP, as increasing volume concentrations increased COP, whereas an increase in speed lowered the COP. Figure 10 shows the variation in the power consumption for different refrigerant charges and speeds, with the volume concentration fixed at 0.025%. An increase in refrigerant charge and speed resulted in significant increments in power consumption.

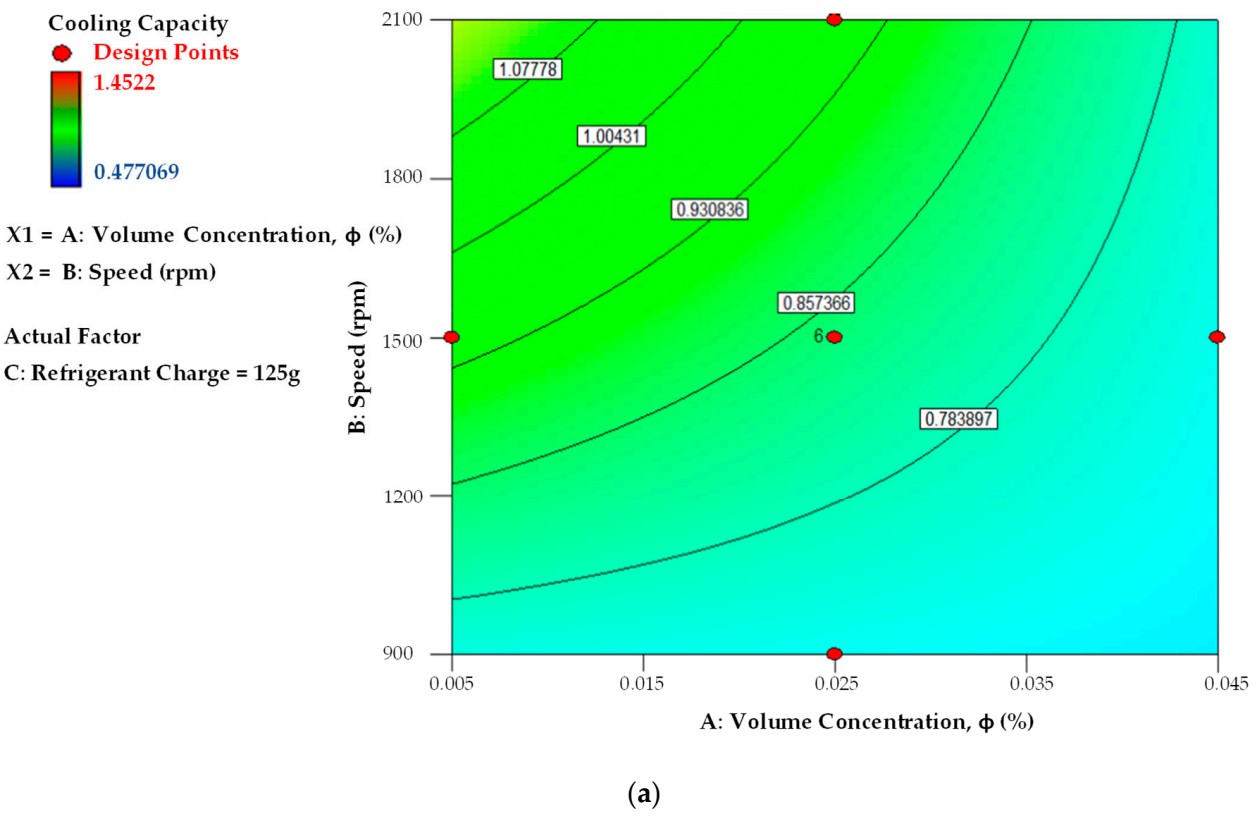

(**a**)

**Figure 7.** *Cont.*

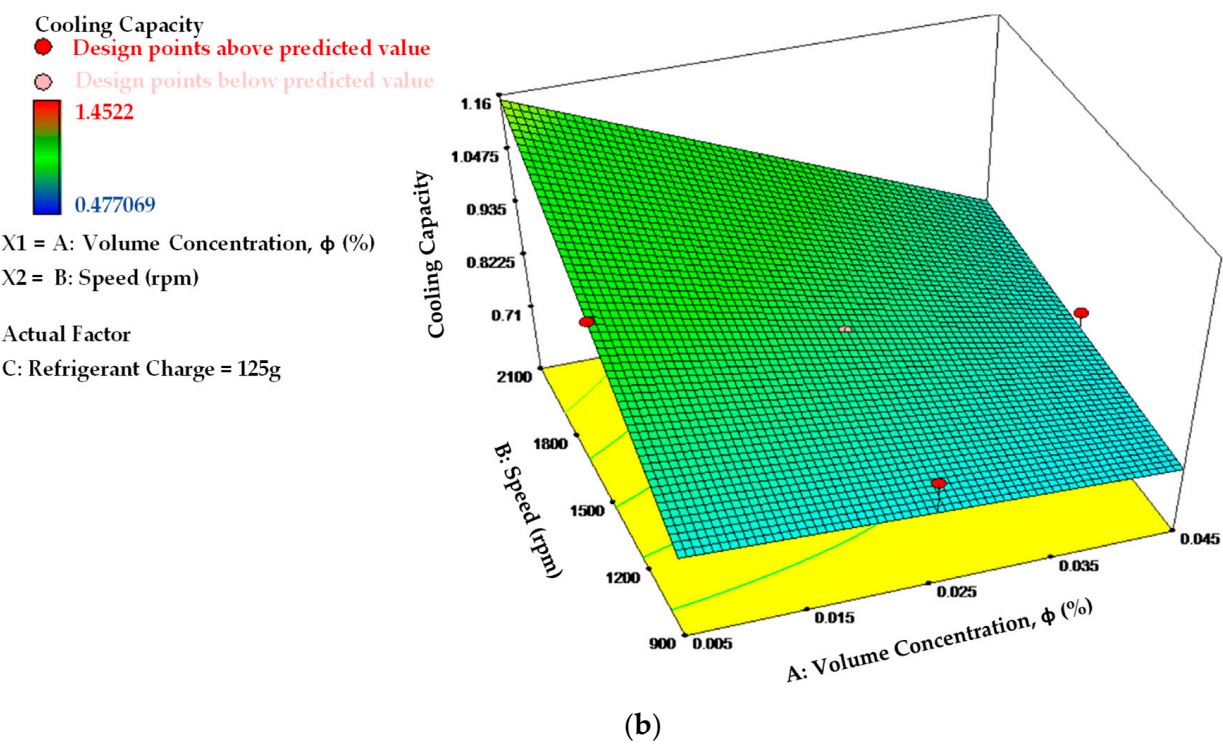

**Figure 7.** Effects of speed and refrigerant charge on cooling capacity: (**a**) contour plot; (**b**) 3D contour plot.

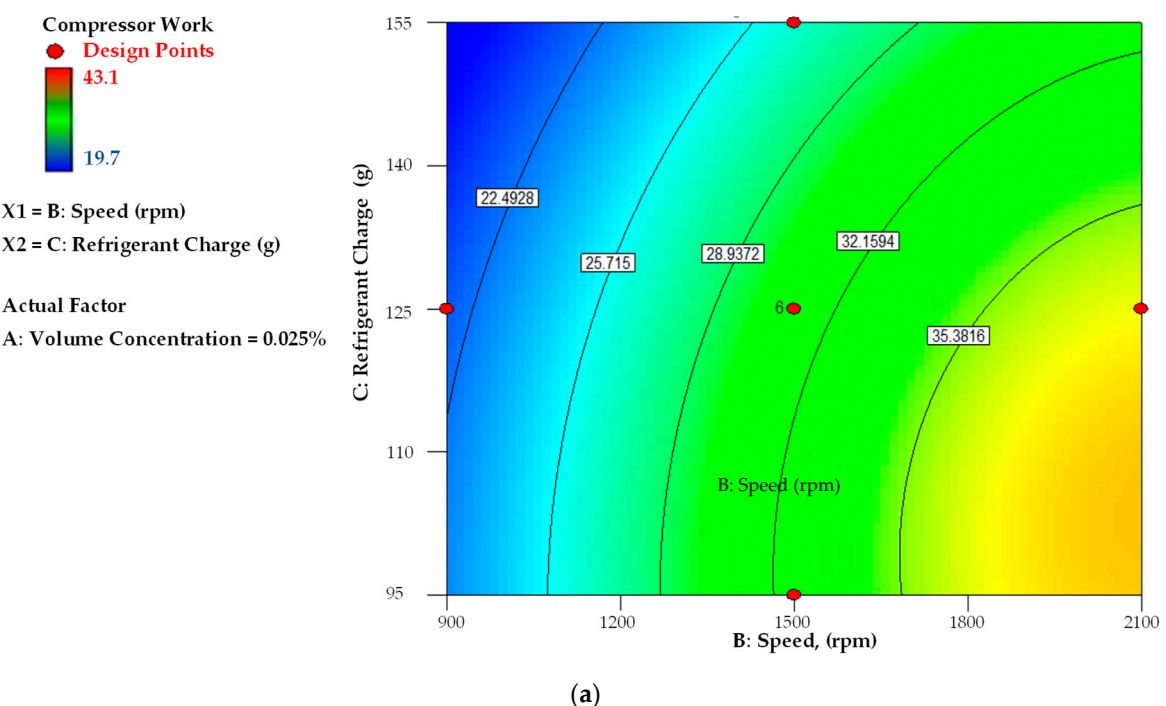

**Figure 8.** *Cont*.

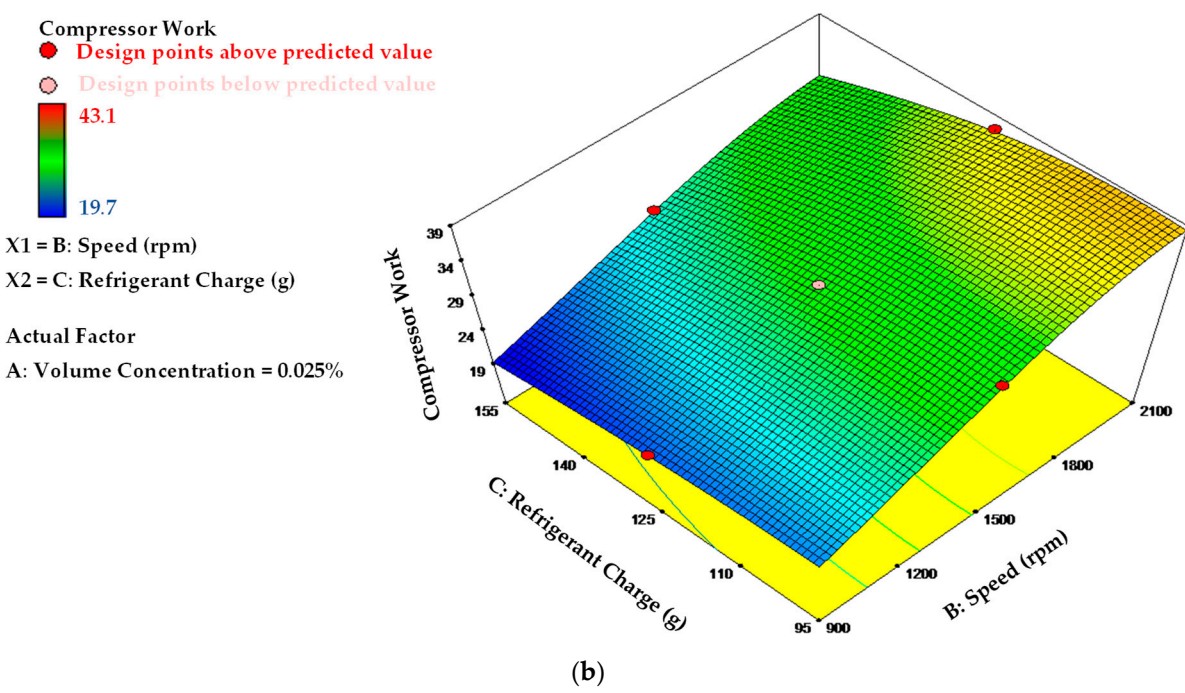

(**b**)

**Figure 8.** Effects of speed and refrigerant charge on compressor work: (**a**) contour plot; (**b**) 3D contour plot.

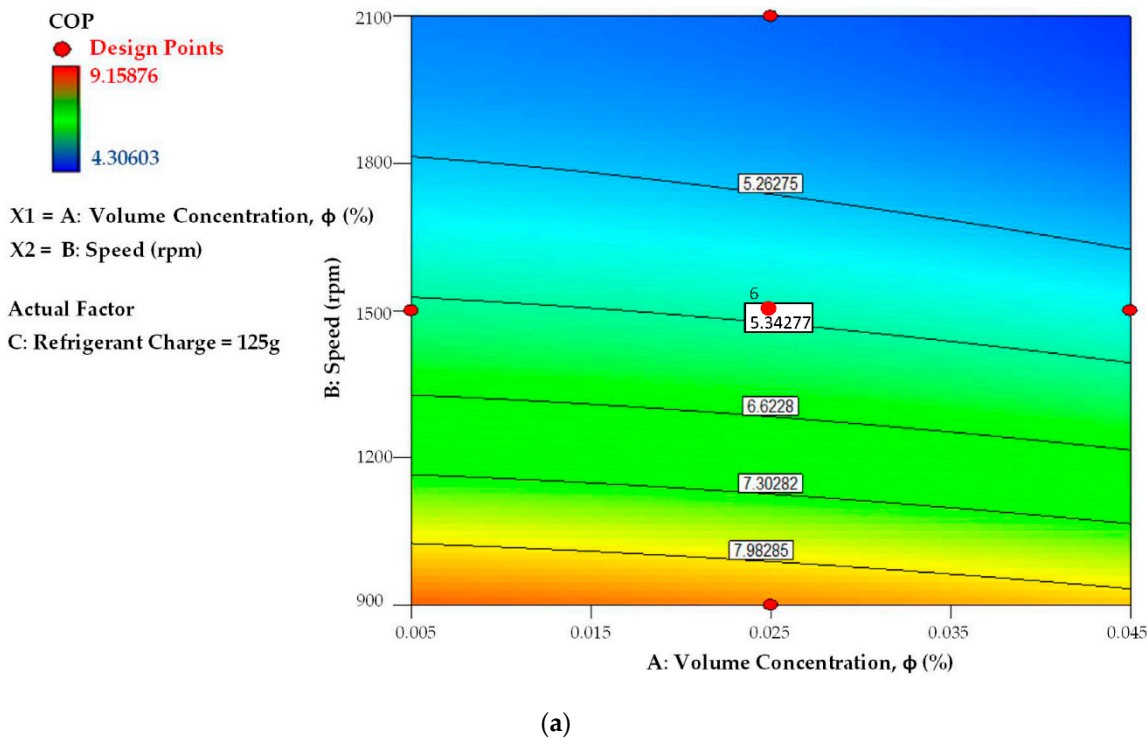

(**a**)

**Figure 9.** *Cont*.

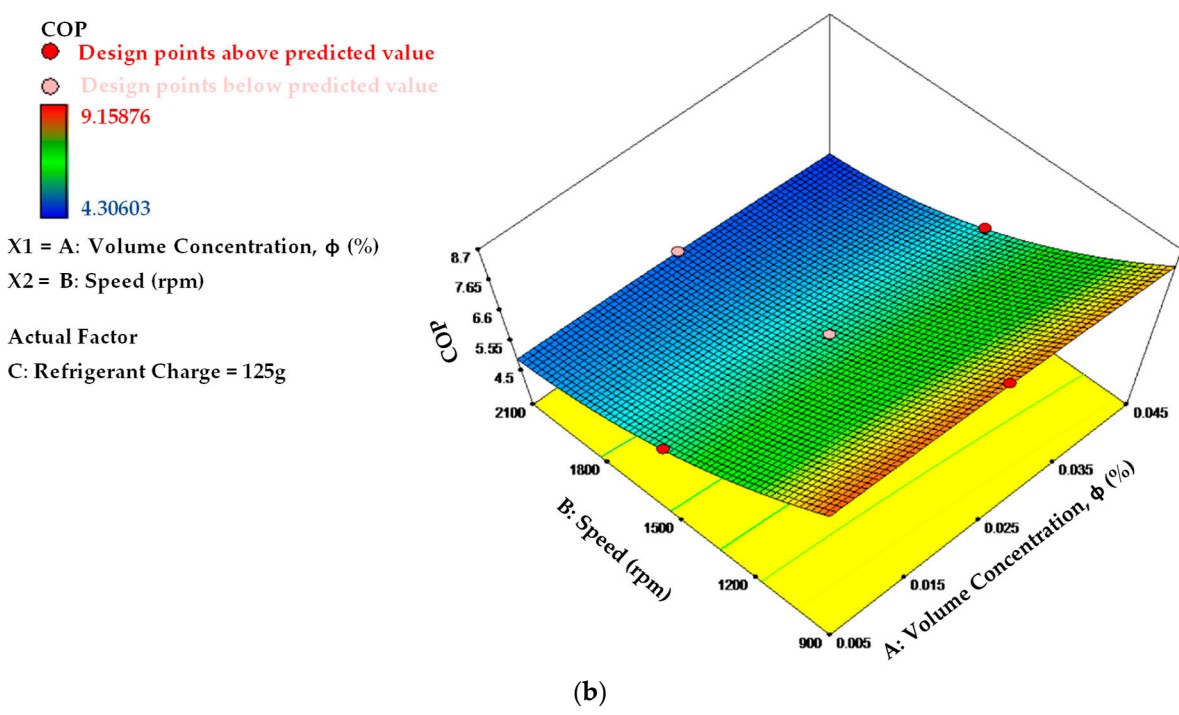

(**b**)

**Figure 9.** Effects of speed and refrigerant charge on COP: (**a**) contour plot; (**b**) 3D contour plot.

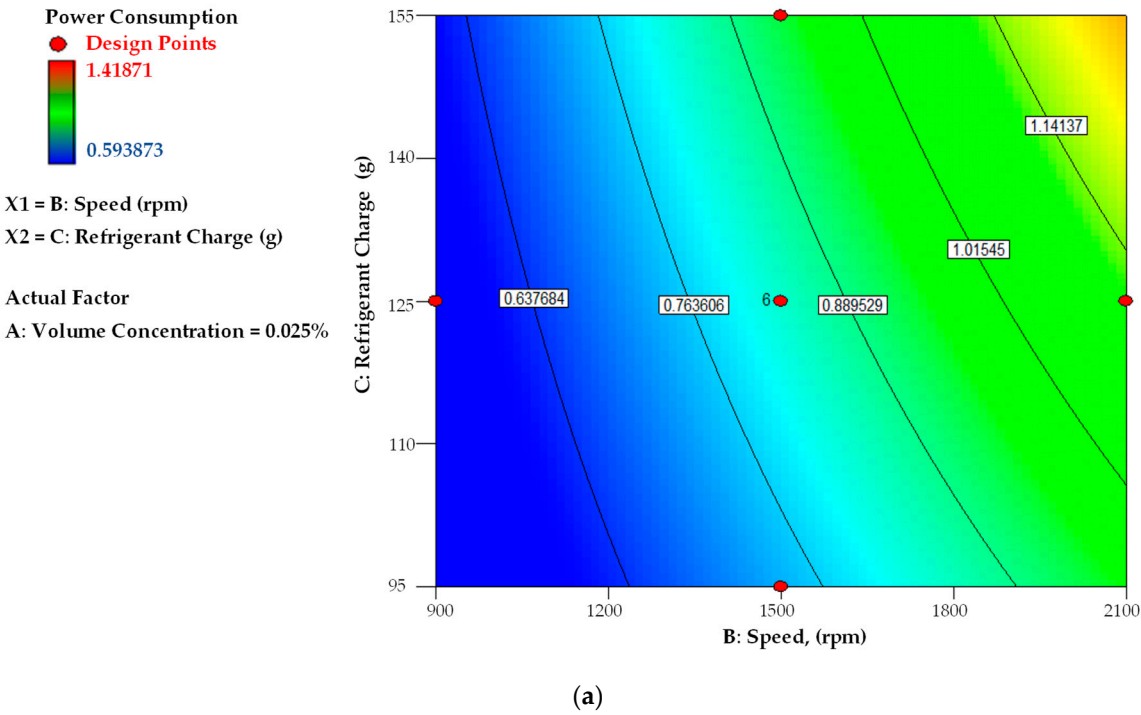

(**a**)

**Figure 10.** *Cont*.

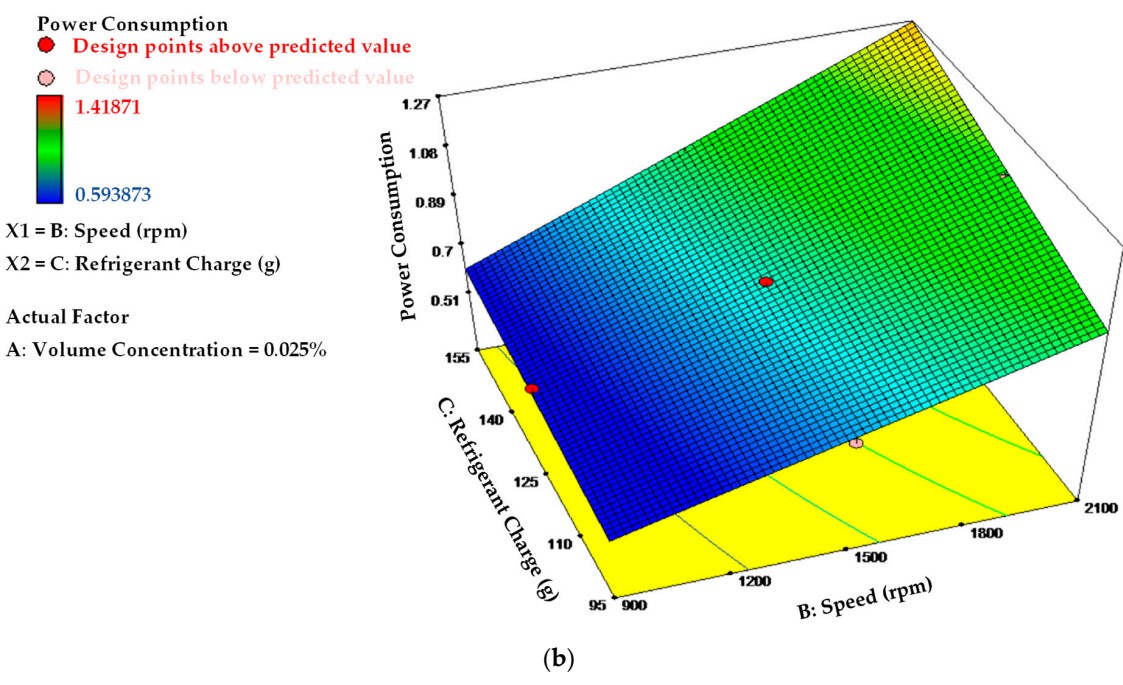

**Figure 10.** Effects of speed and refrigerant charge on power consumption: (**a**) contour plot; (**b**) 3D contour plot.

### 3.4. Optimization and Validation

A confirmation experiment of the control parameters [69] indicated by the RSM optimization technique is required for confirming the improved conditions [70]. Table 14 represents the optimal conditions, with a high desirability of 81.6%. As stated in Table 15, five trial runs at the optimal level were carried out to test and evaluate the reliability of the constructed regression model against the experimental results. The expected and experimental values in the table are quite close to each other. For valid statistical analysis, error values should be less than 20% [71,72]. For all runs, the computed error values were less than 10% and were within acceptable bounds. The validation results were consistent with the current experimental data, reflecting a successful optimization.

**Table 14.** Optimum operating condition.

| Parameter | Optimum Operating Condition |
|---|---|
| A—Volume Concentration, φ (%) | 0.019 |
| B—Compressor Speed (rpm) | 900 |
| C—Initial Refrigerant Charge (g) | 155 |

**Table 15.** Validation Results.

| No. | Responses | | | | | | | | | | | |
|---|---|---|---|---|---|---|---|---|---|---|---|---|
| | Cooling Capacity | | | Compressor Work | | | COP | | | Power Consumption | | |
| | Pred. | Exp. | % | Pred. | Exp. | % | Pred. | Exp. | % | Pred. | Exp. | % |
| 1 | | 0.976 | 4.23 | | 19.6 | 1.89 | | 9.34 | 3.10 | | 0.650 | 4.43 |
| 2 | | 0.897 | 4.23 | | 18.6 | 3.39 | | 9.87 | 8.31 | | 0.655 | 5.23 |
| 3 | 0.935 | 0.868 | 7.73 | 19.23 | 19.9 | 3.37 | 9.05 | 9.19 | 1.52 | 0.621 | 0.656 | 5.33 |
| 4 | | 0.987 | 5.27 | | 19.3 | 0.36 | | 9.47 | 4.44 | | 0.674 | 7.86 |
| 5 | | 0.874 | 6.93 | | 21.3 | 9.72 | | 8.55 | 5.85 | | 0.671 | 7.38 |
| Avg | | | 5.68 | | | 3.74 | | | 4.64 | | | 6.05 |

## 4. Conclusions

The effects of experimental operating conditions, such as by volume concentration of composite nanolubricant, the compressor speed, and refrigerant charge, on cooling capacity, compressor work, COP, and power consumption were assessed. The optimization of operating conditions for an AAC system was performed in the present work using the RSM method. Based on the results of the RSM model, the optimal operation suggested for optimal AAC performance were found at a compressor speed of 900 rpm, refrigerant charge of 155 g, and volume concentration of 0.019%; cooling capacity = 0.9346 kW, compressor work = 19.2296 kJ/kg, COP = 9.051, and power consumption = 0.6209 kW. The validation test runs were carried out to validate predicted results against the experimental results. The developed model shows that the predicted results are in excellent agreement with the experimental results, with an error value of less than 10%. Therefore, it was recommended to use $Al_2O_3$-$SiO_2$/PAG composite nanolubricants with these operating conditions for optimum performance in the AAC system.

**Author Contributions:** Conceptualization, W.H.A.; data curation, N.N.M.Z. and H.M.A.; formal analysis, N.N.M.Z.; investigation, N.N.M.Z., A.A.M.R. and A.I.R.; methodology, W.H.A.; project administration, W.H.A.; software, A.A.M.R. and H.M.A.; supervision, W.H.A.; validation, A.A.M.R. and A.I.R.; visualization, A.I.R. and H.M.A.; writing—original draft, N.N.M.Z.; writing—review and editing, A.I.R. All authors have read and agreed to the published version of the manuscript.

**Funding:** This research was funded by Universiti Malaysia Pahang, grant number RDU213302.

**Institutional Review Board Statement:** Not applicable.

**Informed Consent Statement:** Not applicable.

**Data Availability Statement:** The data that support the findings of this study are available from the corresponding author (W.H. Azmi) upon reasonable request.

**Acknowledgments:** The authors are appreciative for the financial support provided by the Universiti Malaysia Pahang under the International Publication Grant. The authors further acknowledge the contributions of the research teams from the Center for Research in Advanced Fluid and Processes (Pusat Bendalir) and the Advanced Automotive Liquids Laboratory (AALL), who provided valuable insight and expertise for the current study.

**Conflicts of Interest:** The authors declare no conflict of interest.

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
