# Peer review of "Optimization of Air Conditioning Performance with Al2O3-SiO2/PAG Composite Nanolubricants Using the Response Surface Method"

_lubricants, doi:10.3390/lubricants10100243_

Round 1

Reviewer 1 Report

The authors used the response surface method (RSM) to optimize the performance of the air conditioning system with Al2O3-SiO2/PAG composite nanolubricants. The selection of parameters in the RSM study included volume concentrations, compressor speeds and refrigerant charges. Meanwhile, cooling capacity, compressor work, COP and power consumption were also selected for the output responses of the experiment. The relationship model between response and independent factor groups was established and the process was optimized with the model. It is a highlight that the manuscript investigates the effect of Al2O3-SiO2/PAG composite nanolubricants on air conditioning system performance. However, there are some problems with the article.

Recommendations for this manuscript are as follows:

(1)       The font size of the picture axes in the manuscript should be enlarged, and the horizontal and vertical coordinates should be labeled with units. The title of the vertical coordinate of the figs. (2-7) and the comment in the upper right corner of the figs. (4-7) are incomplete.

(2)       The COP on Page 1, CCD on Page 3 and DOE in the table 5 header should be given in full spelling the first time it appears in the manuscript.

(3)       Elemental composition analysis of the particles should be given to identify the gray particles as Al2O3 and the black particles as SiO2 in fig. 1.

(4)       The input conditions and output results of the latter 6 groups of experiments in Table 5 are identical, so why a replicate design

(5)       There are two models suggested in Table 6 and Table 12, which one did the authors choose? What is the basis for the selection?

(6)     The ANOVA responses for COP, compressor work and power consumption have A2, B2 and C2. Why not appear in the ANOVA responses for cooling capacity?

(7)     It is stated: “All values greater than 0.10, on the other hand, imply that the model terms are not significant.”(page 9) But the P-values of A, AB, AC and B2 in Table 13 were all greater than 0.10, so why did the manuscript conclude that: “All terms except AC were all significant”?

(8)       Please explain the meaning of A2 in Eq. 3 and E in Eq. 5

(9)       Figs. (4-7) fix the refrigerant charge and volume concentration respectively. Why not study the effect of refrigerant charge and volume concentration on the AAC system when fixing the compressor speed

(10)   For performance, why not analyze the effect of the other two parameters on the AAC system when fixing three parameters separately? Instead, only one parameter is fixed and the effect of the other two parameters on the AAC system is analyzed.

Based on the above, the manuscript may be accepted for publication after minor revisions.

Author Response

RESPONSE TO REVIEWERS

Paper title “Optimization of Air Conditioning Performance with Al2O3-SiO2/PAG Composite Nanolubricants using Response Surface Method

Resubmitted to Lubricants

Reviewer 1:

Comments and Suggestions for Authors

The authors used the response surface method (RSM) to optimize the performance of the air conditioning system with Al2O3-SiO2/PAG composite nanolubricants. The selection of parameters in the RSM study included volume concentrations, compressor speeds and refrigerant charges. Meanwhile, cooling capacity, compressor work, COP and power consumption were also selected for the output responses of the experiment. The relationship model between response and independent factor groups was established and the process was optimized with the model. It is a highlight that the manuscript investigates the effect of Al2O3-SiO2/PAG composite nanolubricants on air conditioning system performance. However, there are some problems with the article.

Recommendations for this manuscript are as follows:

(1) The font size of the picture axes in the manuscript should be enlarged, and the horizontal and vertical coordinates should be labeled with units. The title of the vertical coordinate of the figs. (2-7) and the comment in the upper right corner of the figs. (4-7) are incomplete.

Answer:

Thank you for the reminder. The graph labelling and font size in the texts were revised accordingly.

(2) The COP on Page 1, CCD on Page 3 and DOE in the table 5 header should be given in full spelling the first time it appears in the manuscript.

Answer:

Thank you for pointing this out. The manuscript was updated to reflect the full name and any first mentions of its abbreviations.

(3) Elemental composition analysis of the particles should be given to identify the gray particles as Al2O3 and the black particles as SiO2 in fig. 1.

Answer:

Thank you for the recommendations. Sigma-Aldrich, St. Louis, Missouri, USA, and Beijing Deke Daojin Science and Technology Co., Ltd., Beijing, China, respectively, were the suppliers for the Al2O3 and SiO2 nanoparticles. To confirm the presence of both nanoparticles, a chemical composition test was necessary. So, using EDX analysis, the elemental composition of the sample including both nanoparticles was identified. The following changes have been made to the manuscript:

“To confirm the existence of the nanoparticles, a chemical composition test was performed. Therefore, the chemical composition of both nanoparticles was assessed by EDX analysis, as shown in Figure 1. In Figures 1(a) and 1(b), respectively, the elemental composition of the materials for Al2O3 and SiO2 nanoparticles was validated.”

(4) The input conditions and output results of the latter 6 groups of experiments in Table 5 are identical, so why a replicate design?

Answer:

Thank you for the comments. The reasons identical latter 6 groups of the experiements in Table 5 are already explained previously in section 2.2 paragraph 2. They are considered as the center points of the design of experiment (DOE) of the work. A three-factor and a three level face-centered cube design (FCD) consisting of twenty experimental runs was employed including six replicates at the center point. The manuscript was updated to provide further reasons. The manuscript was revised accordingly as below:

“The CCD was used to optimise the model, and it worked well for fitting a quadratic surface and for process optimization in general. In this study, FCD is used because there is a common area of interest and operability, and the trials are based on the design matrix. Each parameter has three levels of variation: (i) high (+1), (ii) low (-1) and (iii) centre points (coded as level 0). Six central points, six axial points and eight factorial points were used in this study with alpha, . The  value is denoted as the distance between each axial point and the CCD's centre Redhwan et al. [1]. Multi-objective responses of AAC performance optimization of optimum design with the highest desirability are sought. Three AAC system parameters with their levels using RSM analysis were investigated. Twenty experimental runs, including six replicates at the centre point, were used in a FCD with three factors and three levels. The factor levels of independent variables for AAC system performance are previously mentioned in Table 4. Table 5 illustrates the complete design matrix for the experiments to be conducted, as well as the collected findings, which were analysed using analysis of variance (ANOVA) by Design Expert Software.”

(5) There are two models suggested in Table 6 and Table 12, which one did the authors choose? What is the basis for the selection?

Answer:

Thank you for the comments. The reasons for selecting models in Table 6 and 12 were briefly discussed in the manuscript. The manuscript was updated to provide further reasons. The manuscript was revised accordingly as below:

Table 6 (in page 9)

“CCD module suggested linear and 2FI model to be use for analysis. In order to analyse the cooling capability, linear and two-way interaction (2FI) model were both employed. The model has been improved by the addition of linear and interaction components, as shown by the low p-value (Prob>F). The quadratic model is not suggested for this case. Qubic model was noted as aliased because of aliased terms existance. The qubic model was not suggested due to insufficient running experiments to independently estimate all the terms.” 

Table 12 (in page 12)

“CCD module suggested linear and quadratic model to be use for analysis. Due to their superior accuracy over linear models, quadratic models were chosen. Qubic model was noted as aliased and not suggested due to insufficient running experiments to independently estimate all the terms.”

(6) The ANOVA responses for COP, compressor work and power consumption have A2, B2 and C2. Why not appear in the ANOVA responses for cooling capacity?

Answer:

Thank you for the comment. The sequential sum of squares for the quadratic; A-squared (A2), B-squared (B2) and C-squared (C2) terms is selected for COP, compressor work, and power consumption. The F-value for all these factors evaluate the importance of including quadratic terms in the two-factor interaction (2FI) model. Consequently, a low p-value (Prob>F) indicates that the quadratic terms have enhanced the model. But when it comes to cooling capacity, the quadratic model is not suggested.

(7) It is stated: “All values greater than 0.10, on the other hand, imply that the model terms are not significant.” (Page 9) But the P-values of A, AB, AC and B2 in Table 13 were all greater than 0.10, so why did the manuscript conclude that: “All terms except AC were all significant”?

Answer:

Thank you for the reminder. An error was done during transferring the data. Revision on the statement is done in the manuscript. In accordance with the reviewer’s remark, we corrected the sentence accordingly as below:

“Model terms with P-values less than 0.05 are considered significant. All values greater than 0.10, on the other hand, imply that the model terms are not significant. In this case A, B, C, BC and A2 are significant model terms.”

(8)       Please explain the meaning of A2 in Eq. 3 and E in Eq. 5

Answer:

Thank you for the comments. The model term A-squared in Eq. 3 stands for the quadratic effect terms of the factor A (volume concentration, ɸ (%)). Meanwhile, E in Eq. 5 stands for the E stands for exponential. So, the value "9.041Ε-0.03" is the equivalent of 9.041X10-0.03 in scientific notation. A clearer detonation for the questions was added to the manuscript. The manuscript was revised accordingly as below:

“… where A is the volume concentration of the composite nanolubricants (%), B is the speed (rpm), and C is the refrigerant charge (g).”

(9)       Figs. (4-7) fix the refrigerant charge and volume concentration respectively. Why not study the effect of refrigerant charge and volume concentration on the AAC system when fixing the compressor speed?

Answer:

Thank you for the recommendations. The authors have not studied the effect of refrigerant charge and volume concentration while fixing the compressor speed. This is due to the effect of compressor speed is dominant when observing the effect of parameters towards the AAC system responses i.e cooling capacity, compressor work, COP and also power consumption. Therefore, it is necessary for the compressor speed factor included as variable factor.

(10)   For performance, why not analyze the effect of the other two parameters on the AAC system when fixing three parameters separately? Instead, only one parameter is fixed and the effect of the other two parameters on the AAC system is analyzed.

Answer:

Thank you for the recommendations. The authors only investigated the impact of two parameters with one fixed parameter on the AAC system's performance. The authors vary the impacts of compressor speed with either of two additional factors, such as refrigerant charge or volume concentration. This is because the performance of the AAC system is significantly impacted by compressor speed compared to other two parameters. With variable speed compressor control, as opposed to operation at a single constant speed, a significant reduction in the compressor's workload and power consumption is observed [2]. The authors concluded that the compressor speed factor must be incorporated as a variable factor.

Based on the above, the manuscript may be accepted for publication after minor revisions.

Reviewer 2 Report

The manuscript lacks clarity and originality about the prospective use of composite materials as nanolubricants for refrigeration.

The RSM findings are presented without any explanation as to why the composite performed better. The thermal capacity of Nanolubricants has not been experimentally studied in order to validate the results.

The authors simply mixed the NPs in a ratio of (60-40) and duplicated their prior work in Ref [24] using RSM.

TEM Photograph of Fred The composite NP is deceptive. It is uncertain whether the combination of two NPs may produce novel Nanocomposite materials without chemical modification.

Authors have not published any information regarding the formulation of NL.

There is no information regarding dispersion issues such as UV-Vis or visual photographs.

Did the author believe that zeta potential 61.1 mV denotes steady dispersion? Please provide an illustration and a list of references. " The current absolute zeta potential for 123 Al2O3-SiO2/PAG is as high as 61.1 mV "

I believe it would be fantastic if particle size in suspension could be provided alongside zeta potential and poly dispersity index from DLS.

How the usage of a combination of NPs affects the COP in comparison to the use of a single NP is neither discussed nor supported by science.

Author Response

RESPONSE TO REVIEWERS

Paper title “Optimization of Air Conditioning Performance with Al2O3-SiO2/PAG Composite Nanolubricants using Response Surface Method

Resubmitted to Lubricants

Reviewer 2:

Comments and Suggestions for Authors

The manuscript lacks clarity and originality about the prospective use of composite materials as nanolubricants for refrigeration.

  1. The RSM findings are presented without any explanation as to why the composite performed better. The thermal capacity of Nanolubricants has not been experimentally studied in order to validate the results.

Answer:

Thank you for the comments. According to earlier research studies [2-7] by the authors that is recorded in the literature, composite nanolubricants outperformed single nanolubricants in terms of compressor operation, stability, wear rates, and AAC system performance. Higher thermal conductivity is observed [5], which further supports the comparison between the thermal capacity of composite nanolubricants and single nanolubricants.The manuscript is revised appropriately to address above issues. The manuscript was updated accordingly as below:

A concept of using two or more metal oxide nanoparticles in existing lubricants—known as composite nanolubricants—is adapted due to the limited contribution of single nanolubricants in terms of stability, compressor operations, wear rates, and performance of AAC system. Nanofluids/nanolubricants have distinct thermal physical, tribological properties and performance than base fluids, according to several investigations [8-11]. Previously, studies on the thermal physical, tribological properties, performance and optimization of AAC performance using PAG based single-component nanolubricants with SiO2, Al2O3 and TiO2 metal oxides are available in the literature [12-16]. Zawawi et al. [5] examined the thermal conductivity of single Al2O3 and SiO2 and metal oxide composite nanolubricants. Based on the comparison, metal oxide composite nanolubricants have a substantially higher thermal conductivity than single nanolubricants.

  1. The authors simply mixed the NPs in a ratio of (60-40) and duplicated their prior work in Ref [24] using RSM.

Answer:

Thank you for the comments. The composition ratio of composite nanolubricants which is referred in this work, which is well-established in the literature, has been thoroughly researched by the authors previously in several works. The Al2O3-SiO2/PAG composite nanolubricants in a 60:40 ratio, according to the authors, produces better thermal characteristics [7], tribological behaviour [17], and AAC system performance [3] compared to other combination ratios. The optimum ratio for Al2O3-SiO2/PAG composite nanolubricants is therefore chosen for the current work as a continuation of the prior work. The manuscript was updated to provide further reasons. The manuscript was revised accordingly as below:

Zawawi et al. [5] found that the best combination for both nanolubricants used is 60:40 composition ratios. The Al2O3-SiO2/PAG composite nanolubricants in a 60:40 ratio, according to the authors, produces better thermal characteristics [7], tribological behaviour [17], and AAC system performance [3] compared to other combination ratios. The optimum ratio for Al2O3-SiO2/PAG composite nanolubricants is therefore chosen for the current work as a continuation of the prior work.”  

  1. TEM Photograph of Fred The composite NP is deceptive. It is uncertain whether the combination of two NPs may produce novel Nanocomposite materials without chemical modification.

Answer:

Thank you for the comments. The authors are well aware that nanoparticle agglomeration has a significant impact on stability, which causes a gap between theoretical and experimental results [18]. The dispersion condition of suspended nanoparticles in PAG lubricants is observed in the current work using transmission electron microscopy (TEM). The TEM imaging method was used to verify the two nanoparticles' size, dispersion, and agglomeration after they had been suspended in PAG lubricants and used in other researchers' experiments [19, 20]. The suspension of the Al2O3-SiO2/PAG composite nanolubricants was confirmed to be stable for up to 30 days by the authors in a prior study using visual observation. Morever, It should be noted that no surfactant was utilised in the research. This is due to the fact that the authors have taken into account the drawbacks of using chemical methods to prepare nanolubricants, such as the addition of surfactant, pH adjustment, and surface modification. Consequently, these methods were not taken into account in the current study to avoid any potential repercussions in future research. New information on the TEM imaging are added to the manuscript. The manuscript has been updated as shown below:

TEM evaluation was carried out for the composite composite nanolubricant to observe the colloidal nanoparticle dispersion in nanolubricants. Figure 2 shows a TEM imaging of Al2O3-SiO2/PAG composite nanolubricants. Both nanoparticles were discovered to be spherical. In addition, the graph demonstrates the presence of two groups of nanoparticles with various diameters. Al2O3 nanoparticles are represented by the smaller diameter particles, while SiO2 nanoparticles are represented by the larger diameter particles. The appearance of nanoparticles in grayscale shades may be caused by overlap particles and small aggregation.”

  1. Authors have not published any information regarding the formulation of NL.

Answer:

Thank you for the comments. Similarly, the formulation of nanolubricants are already well-established in literature and all the composite nanolubricants suspension prepared in this work are referred to the previous works. Additional information on the preparation and formulation of composite nanolubricants are added to the manuscript. The manuscript has been updated as shown below:

“The previous literatures had already established the formulation and characterisation of composite nanolubricants. As a result, the preparation and formulation procedures for composite nanolubricants were addressed. In this study, the Al2O3-SiO2/PAG composite nanolubricants were made utilising a two-step procedure, and their stability was then investigated using UV-Vis and zeta potential.”

  1. There is no information regarding dispersion issues such as UV-Vis or visual photographs.

Answer:

Thank you for the comments. The manuscript includes new details on dispersion-related concerns, such as the UV-Vis study and its illustration. To resolve these issues, the manuscript has been appropriately revised. The manuscript has been updated as shown below:

“The prepared composite nanolubricants were then sonicated in ultrasonic bath for 2 hours for a uniform dispersion and stable suspension based on previous works [2, 4-7, 21] and supported in the Figure 3. The absorbance ratio of the mixed nanolubricant dispersions, measured at various sonication durations (0 to 2.0 hours) up to 700 hours, is shown in Figure 3. The graph is used to determine the ideal sonication duration required to preserve the stability of Al2O3-SiO2/PAG composite nanolubricants. With the most stable composite nanolubricants, the absorbance ratio with the highest value indicates the ideal sonication time. According to the graph, two hours of sonication sustained the concentration ratio beyond 90% even after up to 700 hours of sedimentation.”

  1. Did the author believe that zeta potential 61.1 mV denotes steady dispersion? Please provide an illustration and a list of references. " The current absolute zeta potential for Al2O3-SiO2/PAG is as high as 61.1 mV "

Answer:

Thank you for the comments. The authors held the opinion that steady dispersion is indicated by the zeta potential of 61.1 mV. This is corroborated by Lee et al. [22] findings, which showed that a suspension with a zeta potential stability of more than 60 mV was in excellent stability condition. The claim is supported by a zeta potential illustration that contrasts several composite nanolubricants with single nanolubricants with references listed. The manuscript is revised appropriately to address above issues. The manuscript was updated accordingly as below:

“The current absolute zeta potential reading for the Al2O3-SiO2/PAG is up to 61.1 mV whereas other combination of metal oxides i.e Al2O3-TiO2/PAG and TiO2-SiO2/PAG composite nanolubricants which were studied prior to this work [5] recorded up to 31.7 mV and 22.7 mV, respectively. Previously, Redhwan et al. [23] reported that the zeta potential for Al2O3/PAG single nanolubricants was 37.8 mV. When compared to single-component nanolubricants, the Al2O3-SiO2/PAG composite nanolubricants employed in this study showed improved stability. The present results were compared to the stability classification as suggested by Lee et al. [22] as shown in Figure 4. The zeta potential for Al2O3-SiO2/PAG was found to be beyond the stable limit; thus, proving an excellent stability.”

  1. I believe it would be fantastic if particle size in suspension could be provided alongside zeta potential and poly dispersity index from DLS.

Answer:

Thank you for the recommendtions. Zeta potential, poly dispersity index, and more details on the particle size suspension are included in the manuscript. The manuscript was revised accordingly as below:

“The zeta potential and zeta sizer were used in the study to analyse the zeta potential reading and polydispersity index (PDI) of the composite nanolubricants.”

“The breadth or spread of the particle size distribution is described by the PDI, which is another crucial metric [24]. The maximum PDI value was found to be 0.86 for the Al2O3-TiO2/PAG composite nanolubricants, while the lowest PDI value was found to be 0.22 for the Al2O3-SiO2/PAG, as can be seen in Fig. In light of this, it should be observed that the lowest PDI value is quite comparable to the mono-disperse state. A suspension will be mono-disperse, according to Sadeghi et al. [25], if the PDI value is less than 0.3 and the size distribution curve has a single peak.”

  1. How the usage of a combination of NPs affects the COP in comparison to the use of a single NP is neither discussed nor supported by science.

Answer:

Thank you for the comments. The combination of nanoparticles Al2O3-SiO2/PAG does have an influence on the COP of the AAC system. The existing composite nanolubricants are contrasted with individual nanoparticles of each type of nanoparticle that is involved. Comparing composite nanolubricants to SiO2/PAG nanolubricants, it has been found that they increase the system's COP, but they significantly underperform Al2O3/PAG nanolubricants in this regard. However, the composite nanolubricants outperformed Al2O3/PAG significantly in terms of cooling capacity performance. This confirms that composite nanolubricants perform better in terms of the AAC system performance than single nanolubricants. Additional information is added for clarification for abovw issues. The manuscript was revised accordingly as below:

“Additionally,ew studies investigated the performance of single nanolubricants and composite nanolubricants in the refrigeration and AAC system [3, 23, 26, 27]. Sharif et al. [27] examined the performance of the AAC system employing SiO2/PAG nanolubricants. They discovered maximal COP enhancement of up to 24%. In other study, Redhwan et al. [23] claimed that COP and cooling capacity were improved by up to 31% and 32%, respectively, in another experiment. Meanwhile, Zawawi et al. [3] found that Al2O3-SiO2/PAG composite nanolubricants had greater COP and cooling capacity increase than single nanolubricants, with values of 28.10% and 65.21% at 0.015% volume concentration.”

Reference

[1]           Redhwan, A. A. M., Azmi, W. H., Najafi, G., Sharif, M. Z., and Zawawi, N. N. M., 2018, "Application of response surface methodology in optimization of automotive air-conditioning performance operating with SiO2/PAG nanolubricant," Journal of Thermal Analysis and Calorimetry, pp. 1-15.

[2]           Zawawi, N. N. M., and Azmi, W. H., 2020, "Performance of Al2O3-SiO2/PAG employed composite nanolubricant in automotive air conditioning (AAC) system," IOP Conference Series: Materials Science and Engineering, IOP Publishing, p. 012052.

[3]           Zawawi, N. N. M., Azmi, W. H., and Ghazali, M. F., 2022, "Performance of Al2O3-SiO2/PAG composite nanolubricants in automotive air-conditioning system," Applied Thermal Engineering, 204, p. 117998.

[4]           Zawawi, N. N. M., Azmi, W. H., Redhwan, A. A. M., and Sharif, M. Z., 2018, "Thermo-physical properties of metal oxides composite Nanolubricants," Journal of Mechanical Engineering, 15(1), pp. 28-38.

[5]           Zawawi, N. N. M., Azmi, W. H., Redhwan, A. A. M., Sharif, M. Z., and Samykano, M., 2018, "Experimental investigation on thermo-physical properties of metal oxide composite nanolubricants," International Journal of Refrigeration, 89, pp. 11-21.

[6]           Zawawi, N. N. M., Azmi, W. H., Redhwan, A. A. M., Sharif, M. Z., and Sharma, K. V., 2017, "Thermo-physical properties of Al2O3-SiO2/PAG composite nanolubricant for refrigeration system," International Journal of Refrigeration, 80, pp. 1-10.

[7]           Zawawi, N. N. M., Azmi, W. H., Sharif, M. Z., and Najafi, G., 2019, "Experimental investigation on stability and thermo-physical properties of Al2O3–SiO2/PAG nanolubricants with different nanoparticle ratios," Journal of Thermal Analysis and Calorimetry, 135(2), pp. 1243-1255.

[8]           Bhiradi, I., and Hiremath, S. S., 2020, "Energy efficient and cost effective method for generation of in-situ silver nanofluids: Formation, morphology and thermal properties," Advanced Powder Technology, 31(9), pp. 4031-4044.

[9]           Ying, Z., He, B., He, D., Kuang, Y., Ren, J., and Song, B., 2020, "Comparisons of single-phase and two-phase models for numerical predictions of Al2O3/water nanofluids convective heat transfer," Advanced Powder Technology, 31(7), pp. 3050-3061.

[10]         Singh, S. K., and Sarkar, J., 2020, "Improving hydrothermal performance of hybrid nanofluid in double tube heat exchanger using tapered wire coil turbulator," Advanced Powder Technology, 31(5), pp. 2092-2100.

[11]         Anitha, S., Thomas, T., Parthiban, V., and Pichumani, M., 2019, "What dominates heat transfer performance of hybrid nanofluid in single pass shell and tube heat exchanger?," Advanced Powder Technology, 30(12), pp. 3107-3117.

[12]         Sharif, M. Z., Azmi, W. H., Redhwan, A. A. M., and Mamat, R., 2016, "Investigation of thermal conductivity and viscosity of Al2O3/PAG nanolubricant for application in automotive air conditioning system," international journal of refrigeration, 70, pp. 93-102.

[13]         Redhwan, A. A. M., Azmi, W. H., Sharif, M. Z., and Mamat, R., 2017, "Comparative study of thermo-physical properties of SiO2 and Al2O3 nanoparticles dispersed in PAG lubricant," Applied Thermal Engineering, 116, pp. 823-832.

[14]         Redhwan, A., Azmi, W., and Sharif, M., 2017, "Thermal conductivity enhancement of Al2O3 and SiO2 nanolubricants for application in automotive air conditioning (AAC) system," MATEC Web of Conferences, EDP Sciences, p. 01051.

[15]         Sanukrishna, S., and Prakash, M. J., 2018, "Experimental studies on thermal and rheological behaviour of TiO2-PAG nanolubricant for refrigeration system," International Journal of Refrigeration, 86, pp. 356-372.

[16]         Sanukrishna, S., Vishnu, S., and Prakash, M. J., 2018, "Experimental investigation on thermal and rheological behaviour of PAG lubricant modified with SiO2 nanoparticles," Journal of Molecular Liquids, 261, pp. 411-422.

[17]         Zawawi, N., Azmi, W., and Ghazali, M., 2022, "Tribological performance of Al2O3–SiO2/PAG composite nanolubricants for application in air-conditioning compressor," Wear, p. 204238.

[18]         Bi, S., Guo, K., Liu, Z., and Wu, J., 2011, "Performance of a domestic refrigerator using TiO2-R600a nano-refrigerant as working fluid," Energy Conversion and Management, 52(1), pp. 733-737.

[19]         Sundar, L. S., Irurueta, G. O., Venkata Ramana, E., Singh, M. K., and Sousa, A. C. M., 2016, "Thermal conductivity and viscosity of hybrid nanfluids prepared with magnetic nanodiamond-cobalt oxide (ND-Co3O4) nanocomposite," Case Studies in Thermal Engineering, 7, pp. 66-77.

[20]         Mohammed, H. I., Giddings, D., and Walker, G. S., 2020, "Thermo-physical properties of the nano-binary fluid (acetone–zinc bromide-ZnO) as a low temperature operating fluid for use in an absorption refrigeration machine," Heat and Mass Transfer, 56(3), pp. 1037-1044.

[21]         Zawawi, N. N. M., Azmi, W. H., Sharif, M. Z., and Shaiful, A. I. M., 2019, "Composite nanolubricants in automotive air conditioning system: An investigation on its performance," IOP Conference Series: Materials Science and Engineering, IOP Publishing, p. 012078.

[22]         Lee, J. H., Hwang, K. S., Jang, S. P., Lee, B. H., Kim, J. H., Choi, S. U. S., and Choi, C. J., 2008, "Effective viscosities and thermal conductivities of aqueous nanofluids containing low volume concentrations of Al2O3 nanoparticles," International Journal of Heat and Mass Transfer, 51(11–12), pp. 2651-2656.

[23]         Redhwan, A. A. M., Azmi, W. H., Sharif, M. Z., Mamat, R., Samykano, M., and Najafi, G., 2019, "Performance improvement in mobile air conditioning system using Al2O3/PAG nanolubricant," Journal of Thermal Analysis and Calorimetry, 135(2), pp. 1299-1310.

[24]         Raval, N., Maheshwari, R., Kalyane, D., Youngren-Ortiz, S. R., Chougule, M. B., and Tekade, R. K., 2019, "Importance of physicochemical characterization of nanoparticles in pharmaceutical product development," Basic fundamentals of drug delivery, Elsevier, pp. 369-400.

[25]         Sadeghi, R., Etemad, S. G., Keshavarzi, E., and Haghshenasfard, M., 2015, "Investigation of alumina nanofluid stability by UV–vis spectrum," Microfluidics and Nanofluidics, 18(5), pp. 1023-1030.

[26]         Sanukrishna, S. S., and Prakash, M. J., 2018, "Experimental studies on thermal and rheological behaviour of TiO2-PAG nanolubricant for refrigeration system," International Journal of Refrigeration, 86, pp. 356-372.

[27]         Sharif, M. Z., Azmi, W. H., Redhwan, A. A. M., Mamat, R., and Yusof, T. M., 2017, "Performance analysis of SiO2/PAG nanolubricant in automotive air conditioning system," International Journal of Refrigeration, 75, pp. 204-216.

Round 2

Reviewer 2 Report

The authors have incorporated most of the Queries.

Although authors have self-cited their work unnecessarily. Please avoid the usage of clubbing and use only desired citations.

[46,56-58,60] was used just to cite "Properties of Nanoparticles"

Author Response

RESPONSE TO REVIEWERS

Paper title “Optimization of Air Conditioning Performance with Al2O3-SiO2/PAG Composite Nanolubricants using Response Surface Method

Resubmitted to Lubricants

Reviewer:

Comments and Suggestions for Authors

The authors have incorporated most of the Queries.

Answer:

Thank you for the comment.

Although authors have self-cited their work unnecessarily. Please avoid the usage of clubbing and use only desired citations.

[46,56-58,60] was used just to cite "Properties of Nanoparticles"

Answer:

Thank you for the reminder. Unnecessary self-citations are removed, the usage of clubbing in the references is avoided, and only desired citations are retained. The manuscript was updated to reflect the revisions. The following changes have been made to the manuscript:

Page 4 Paragraph 2

“Additionally, few studies investigated the performance of single nanolubricants and composite nanolubricants in the refrigeration and AAC system [41,52,53].”

“The literature on the use of composite nanolubricants to improve the performance of AAC systems is scarce [55].”

Page 5 Paragraph 1

“Table 2 lists the features of these nanoparticles [46,56] and Table 3 illustrates the char-acteristics of PAG 46 lubricant at atmospheric pressure [57].”

The references for Table 2 and 3 are also updated following the recent revisions. The following changes have been made to the manuscript:

“Table 2. Properties of Nanoparticles [46,56]”

“Table 3. Properties of PAG 46 lubricant [57]”